# LOGARITHMIC UNBIASED QUANTIZATION: PRACTICAL 4-BIT TRAINING IN DEEP LEARNING

## ABSTRACT

Quantization of the weights and activations is one of the main methods to reduce the computational footprint of Deep Neural Networks (DNNs) training. Current methods enable 4-bit quantization of the forward phase. However, this constitutes only a third of the training process. Reducing the computational footprint of the entire training process requires the quantization of the neural gradients, i.e., the loss gradients with respect to the outputs of intermediate neural layers. In this work, we examine the importance of having unbiased quantization in quantized neural network training, where to maintain it, and how. Based on this, we suggest a *logarithmic unbiased quantization* (LUQ) method to quantize both the forward and backward phase to 4-bit, achieving state-of-the-art results in 4-bit training. For example, in ResNet50 on ImageNet, we achieved a degradation of 1.18%; we further improve this to degradation of only 0.64% after a single epoch of high precision fine-tuning combined with a variance reduction method. Finally, we suggest a method that exploits the low precision format by avoiding multiplications during two-thirds of the training process, thus reducing by 5x the area used by the multiplier. A reference implementation is supplied in the supplementary material.

## 1 INTRODUCTION

Deep neural networks (DNNs) are a powerful tool that has shown superior performance in various tasks spanning computer vision, natural language processing, autonomous cars, and more. Unfortunately, their vast demand for computational resources, especially during the training process, is one of the main bottlenecks in the evolution of these models.

Training of DNNs consists of three main general-matrix-multiply (GEMM) phases: the forward phase, backward phase, and update phase. Quantization has become one of the main methods to compress DNNs and reduce the GEMM computational resources. There has been a significant advance in the quantization of the forward phase. There, it is possible to quantize the weights and activations to 4 bits while preserving model accuracy (Banner et al., 2019; Nahshan et al., 2019; Bhalgat et al., 2020; Choi et al., 2018b). Despite these advances, they only apply to a third of the training process while the backward phase and update phase are still computed with higher precision.

Recently, Sun et al. (2020) was able, for the first time, to train a DNN while reducing the numerical precision of most of its parts to 4 bits with some degradation (e.g., 2.49% error in ResNet50). To do so, Sun et al. (2020) suggested a non-standard radix-4 floating-point format, combined with double quantization of the neural gradients (called two-phase rounding). This was an impressive step forward in the ability to quantize all the training processes. However, since a radix-4 format is not aligned with conventional radix-2, any numerical conversion between the two requires an explicit multiplication to modify both the exponent and mantissa and may require additional hardware (Kupriianova et al., 2013). Thus, their non-standard quantization requires specific hardware support that can reduce the benefit of quantization to low bits.

The main challenge in reducing the numerical precision of the entire training process is quantizing the neural gradients, i.e. the backpropagated error. Specifically, Chmiel et al. (2021) showed the neural gradients have a heavy tailed near-lognormal distribution, and therefore they should be logarithmically quantized at low precision levels. For example, for FP4 the optimal format was [sign,exponent,mantissa] = [1,3,0], i.e. without mantissa bits. In contrast, that weights and activations

are well approximated with Normal or Laplacian distributions (Banner et al., 2019; Choi et al., 2018a), and therefore are better approximated using uniform quantization (e.g., INT4).

In this work, in order to reduce the computational resources bottleneck, we dive deeper into understanding neural gradients' quantization. Based on the findings of Chmiel et al. (2021), we focus on the [1,3,0] FP4 format for the neural gradients. We analyze different types of quantization stochastic rounding schemes to explain the importance of having an unbiased rounding scheme for the neural gradients with such a logarithmic quantization format. We compare it with forward phase quantization, where the bias is not a critical property.

Building on this analysis, we suggest a method called *logarithmic unbiased quantization* (LUQ) for unbiased quantization of the neural gradients to the standard FP4 format of [1,3,0]. This, together with quantization of the forward phase to INT4, achieves state-of-the-art results in full training in 4 bits (e.g., 1.18% error in ResNet50), with no overhead. Moreover, we suggest two additional methods to further reduce the degradation, with some overhead: the first method reduces the quantization variance of the neural gradients, while the second is a simple method of fine-tuning in high precision. Combining LUQ with these two proposed methods we achieve, for the first time, only 0.64% error in 4-bit training of ResNet50. The overhead of our additional methods is no more than the previously presented in Sun et al. (2020).

Finally, we exploit the specific FP4 format mantissa used for the neural gradients ([1,3,0]) and suggest replacing the multiplication blocks with a proposed *multiplication free backpropagation* (MF-BPROP) block. This way we completely avoid multiplication in the backward phase and the update phase (which constitute two thirds of the training) and reduce by 5x the area which was previously used for the multiplications in these phases.

The main contributions of this paper:

- A comparison of different rounding schemes.
- We suggest a simple and hardware friendly logarithmically unbiased quantization for the neural gradients called LUQ.
- We demonstrate that two simple methods can further improve the accuracy in 4-bit training: (1) variance reduction using re-sampling and (2) high precision fine-tuning for one epoch.
- We design a modern hardware block that exploits LUQ quantization to avoid multiplication in two thirds of the training process, thus reducing by 5x the multiplier logical area.

## 2 ROUNDING SCHEMES COMPARISON

In this section, we study the effects of unbiased rounding for the forward and backward passes. We show that rounding-to-nearest (RDN) should be applied for the forward phase while stochastic rounding (SR) is more suitable for the backward phase. Specifically, We show that although SR is unbiased, it generically has worse mean-square-error compared to RDN.

Given that we want to quantize $x$ in a bin with a lower limit $l(x)$ and an upper limit $u(x)$, stochastic rounding can be stated as follows:

$$\text{SR}(x) = \begin{cases} l(x), & \text{with probability } p(x) = 1 - \frac{x-l(x)}{u(x)-l(x)} \\ u(x), & \text{with probability } 1 - p(x) = \frac{x-l(x)}{u(x)-l(x)} \end{cases} . \tag{1}$$

The expected rounding value is given by

$$E[\text{SR}(x)] = l(x) \cdot p(x) + u(x) \cdot (1 - p(x)) = x, \tag{2}$$

where here and below the expectation is over the randomness of SR (i.e., $x$ is a deterministic constant). Therefore, stochastic rounding is an unbiased approximation of $x$, since it has zero bias:

$$\text{Bias}[\text{SR}(\text{x})] = E[\text{SR}(x) - x] = E[\text{SR}(x)] - x = 0. \tag{3}$$

However, stochastic rounding has variance, given by

$$\begin{aligned} \text{Var}[\text{SR}(x)] &= (l(x) - E[\text{SR}(x)])^2 \cdot p(x) + (u(x) - E[\text{SR}(x)])^2 \cdot (1 - p(x)) \\ &= (x - l(x)) \cdot (u(x) - x), \end{aligned} \tag{4}$$

where the last transition follows from substituting the terms $E[\mathrm{SR}(x)])$, and $p(x)$ into Eq. (4).

We turn to consider the round-to-nearest method (RDN). The bias of RDN is given by

$$\mathrm{Bias}[\mathrm{RDN(x)}] = \min\left(x - l(x), u(x) - x\right). \tag{5}$$

Since RDN is a deterministic method, it is evident that the variance is 0 i.e.,

$$\mathrm{Var}[\mathrm{RDN(x)}] = 0. \tag{6}$$

Finally for every value $x$ and a rounding method $R(x)$, the mean-square-error (MSE) can be written as the sum of the rounding variance and the squared rounding bias,

$$\mathrm{MSE}[\mathrm{R(x)}] = E[R(x) - x]^2 = \mathrm{Var}[\mathrm{R(x)}] + \mathrm{Bias}^2[\mathrm{R(x)}]. \tag{7}$$

Therefore, we have the following MSE distortion when using round-to-nearest and stochastic rounding:

$$\mathrm{MSE} = \begin{cases} [\min\left(x - l(x), u(x) - x\right)]^2 & RDN(x) \\ (x - l(x)) \cdot (u(x) - x) & SR(x) \end{cases}. \tag{8}$$

Note that since $\min(a,b)^2 \leq a \cdot b$ for every $a, b$, we have that

$$\mathrm{MSE}\left[SR\left(x\right)\right] \geq \mathrm{MSE}[RDN(x)], \quad \forall x. \tag{9}$$

In Fig. 1a we plot the mean-square-error for $x \in [0, 1]$, $l(x) = 0$, and $u(x) = 1$. While round-to-nearest has a lower MSE than stochastic rounding, the former is a biased estimator.

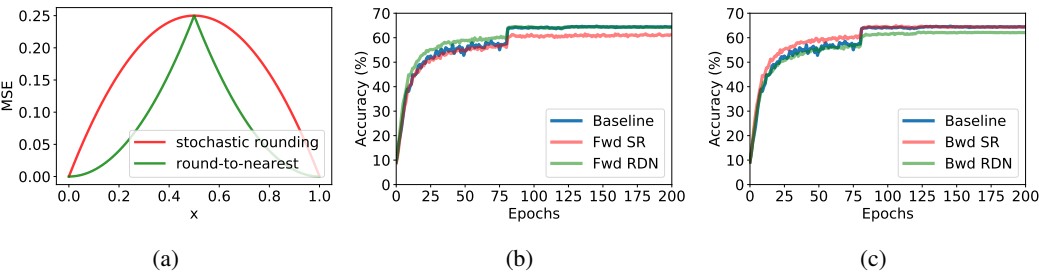

|     |     |     |
| :-: | :-: | :-: |
| (a) | (b) | (c) |

Figure 1: Comparison between stochastic rounding (SR) and round-to-nearest (RDN) quantization. In **(a)** we present the MSE of a uniform distributed tensor with the two different rounding schemes. Quantization to 4 bits of the forward phase **(b)** and backward phase **(c)** of ResNet18 - Cifar100 dataset with SR and RDN. Notice that while MSE is important in the forward phase, unbiasness achieved with SR is crucial for the backward phase. The bwd and fwd in (b) and (c) respectively, are in full precision to focus on the effect of the rounding scheme only in one pass of the network in each experiment.

## 2.1 BACKGROUND: UNBIASED GRADIENT ESTIMATES

To prove convergence, textbook analyses of SGD typically assume the expectation of the (mini-batch) weight gradients is sufficiently close to the true (full-batch) gradient (e.g., assumption 4.3 in (Bottou et al., 2018)). This assumption is satisfied when the weight gradients are unbiased. Next, we show that this condition is met when the neural gradients are quantized without bias.

Denote $W_l$ as the weights between layer $l - 1$ and $l$, $C$ the cost function, and $f_l$ as the activation function at layer $l$. Given an input–output pair $(x, y)$, the loss is:

$$C\left(y, f_L\left(W_L f_{L-1}\left(W_{L-1} \cdots f_2\left(W_2 f_1\left(W_1 x\right)\right) \cdots\right)\right)\right). \tag{10}$$

Let $z^l$ be the weighted input (pre-activation) of layer $l$ and denote the output (activation) of layer $l$ by $a_l$. The derivative of the loss in terms of the inputs is given by the chain rule:

$$\delta_l = \frac{dC}{da_L} \cdot \frac{da_L}{dz_L} \cdot \frac{dz_L}{da_{L-1}} \cdot \frac{da_{L-1}}{dz_{L-1}} \cdot \frac{dz_{L-1}}{da_{L-2}} \cdots \frac{da_l}{dz_l} \cdot \frac{dz_l}{da_{l-1}}. \tag{11}$$

In its quantized version, Eq. (11) gets the following form.

$$\delta_{l_q} = Q\left(\frac{dC}{da_L}\right) \cdot Q\left(\frac{da_L}{dz_L}\right) \cdots Q\left(\frac{da_1}{dz_1}\right) \cdot Q\left(\frac{dz_1}{da_{l-1}}\right) . \tag{12}$$

Assuming $Q(x)$ is an unbiased stochastic quantizer with $E[Q(x)] = x$, the quantized backpropogation $\delta_{l_q}$ is an unbiased approximation of backpropogation $\delta_l$

$$
\begin{aligned}
E[\delta_{l_q}] &= E\left[Q\left(\frac{dC}{da_L}\right) \cdot Q\left(\frac{da_L}{dz_L}\right) \cdots Q\left(\frac{da_1}{dz_1}\right) \cdot Q\left(\frac{dz_1}{da_{l-1}}\right)\right] \\
&= E\left[Q\left(\frac{dC}{da_L}\right)\right] \cdot E\left[Q\left(\frac{da_L}{dz_L}\right)\right] \cdots E\left[Q\left(\frac{da_1}{dz_1}\right)\right] \cdot E\left[Q\left(\frac{dz_1}{da_{l-1}}\right)\right] \\
&= \frac{dC}{da_L} \cdot \frac{da_L}{dz_L} \cdot \frac{dz_L}{da_{L-1}} \cdot \frac{da_{L-1}}{dz_{L-1}} \cdot \frac{dz_{L-1}}{da_{L-2}} \cdots \frac{da_1}{dz_1} \cdot \frac{dz_1}{da_{l-1}} \\
&= \delta_l
\end{aligned}
\tag{13}
$$

In Eq. (13) we used the *linearity* of back-propagation to express the expected product as a product of expectations. Finally, since the gradient of the weights in layer $l$ is $\nabla_{W_l} C = \delta_l \cdot a_{l-1}$ and in its quantized form it becomes $\nabla_{W_l} C_q = \delta_{l_q} \cdot a_{l-1}$, the update $\nabla_{W_l} C_q$ is an unbiased estimator of $\nabla_{W_l} C$:

$$E\left[\nabla_{W_l} C_q\right] = E\left[\delta_{l_q} \cdot a_{l-1}\right] = E\left[\delta_{l_q}\right] \cdot a_{l-1} = \delta_l \cdot a_{l-1} = \nabla_{W_l} C . \tag{14}$$

The forward pass is different from the backward pass in that unbiasedness at the tensor level is not necessarily a guarantee of unbiasedness at the model level since the activation functions and loss function are not linear. Therefore, even after stochastic quantization, the forward phase remains biased.

**Conclusions.** It was previously proved that unbiased neural gradients quantization leads to an unbiased estimate of the weight gradients (e.g., Chen et al. (2020a)), which enables proper convergence of SGD (Bottou et al., 2018). Thus, bias in the gradients can hurt the performance and should be avoided, even at the cost of increasing the MSE. Therefore, neural gradients, following should be quantized using SR, following subsection 2.1. However, the forward phase should be quantized deterministically (using RDN) since stochastic rounding will not make the loss estimate unbiased (due to the non-linearity of the loss and activation functions) while unnecessarily increasing the MSE (as shown in Eq. (9)). There are cases where adding limited noise, such as dropout, increases MSE but improves generalization. However, this is typically not the case, especially if the noise is large. Figs. 1b and 1c show that these theoretical observations are consistent with empirical observations favoring RDN for the forward pass and SR for the backward pass.

## 3 LUQ - A LOGARITHMIC UNBIASED QUANTIZER

A recent work (Chmiel et al., 2021) showed that the neural gradients can be approximated with the lognormal distribution. This distribution has many values concentrated around the mean but is also heavy-tailed, making the extreme values orders of magnitudes larger than the small values sampled from this distribution. They exploit this fact and showed that the neural gradients can be pruned to a high pruning ratio without accuracy degradation (e.g., 85% in ResNet18 ImageNet dataset), using an unbiased pruning method. We build on top of this pruning method, and combine it with an unbiased logarithmic quantizer, as described below.

**Unbiased stochastic pruning** Given an underflow threshold $\alpha$ we define a stochastic pruning operator, which prunes a given value $x$, as

$$T_\alpha(x) = \begin{cases} x & \text{, if } |x| \geq \alpha \\ \text{sign}(x) \cdot \alpha & w.p. \ \frac{|x|}{\alpha}, \text{if } |x| < \alpha \\ 0 & w.p. \ 1 - \frac{|x|}{\alpha}, \text{if } |x| < \alpha . \end{cases} \tag{15}$$

**Unbiased FP quantizer**   Given an underflow threshold $\alpha$, let $Q_\alpha(X)$ be a FP round-to-nearest $b$-bits quantizer with bins $\{\alpha, 2\alpha, ..., 2^{b-1}\alpha\}$. Assume, without loss of generality, $2^{n-1}\alpha < x < 2^n\alpha$ ($n \in \{0, 1..., b-1\}$) . We will use the following unbiased quantizer, which is a special case of SR (Eq. (1)):

$$Q_\alpha(x) = \begin{cases} 2^{n-1}\alpha & w.p. \ \frac{2^n\alpha-x}{2^n\alpha-2^{n-1}\alpha} \\ 2^n\alpha & w.p. \ 1 - \frac{2^n\alpha-x}{2^n\alpha-2^{n-1}\alpha} = \frac{x-2^{n-1}\alpha}{2^{n-1}\alpha} \ . \end{cases} \tag{16}$$

It is unbiased since

$$E[Q_\alpha(x)] = 2^{n-1}\alpha \cdot \frac{2^n\alpha - x}{2^n\alpha - 2^{n-1}\alpha} + 2^n\alpha \cdot \frac{x - 2^{n-1}\alpha}{2^{n-1}\alpha} = x \,, \tag{17}$$

and as a special case of Eq. (2).

**Underflow threshold**   In order to create an unbiased quantizer, the largest quantization value $2^{b-1}\alpha$ should avoid clipping any values of $x$, otherwise this will create a bias. Therefore, the underflow threshold $\alpha$ is chosen as the optimal unbiased value, i.e

$$\alpha = \frac{\max(|x|)}{2^{2^{b-1}}} \,,$$

where $b = 3$ for FP4.

**Logarithmic rounding**   Traditionally, stochastic rounding as in eq. 16 is implemented by adding a uniform random noise $\epsilon \sim U[-\frac{2^{n-1}\alpha}{2}, \frac{2^{n-1}\alpha}{2}]$ to $x$ and then use a round-to-nearest operation. In order to implement round-to-nearest directly on the exponent, we need to correct an inherent bias since $\alpha \cdot 2^{\lfloor \log(\frac{|x|}{\alpha})\rceil} \neq \alpha \cdot \lfloor 2^{\log(\frac{|x|}{\alpha})} \rceil$.

For a bin $[2^{n-1}, 2^n]$, the midpoint $x_m$ is

$$x_m = \frac{2^n + 2^{n-1}}{2} = \frac{3}{4} \cdot 2^{n-1} \,. \tag{18}$$

Therefore, we can apply round-to-nearest-power (RDNP) directly on the exponent $x$ of any value $2^{n-1} \leq 2^x \leq 2^n$ as follows:

$$\text{RDNP}(2^x) = 2^{\lfloor \log\left(\frac{4}{3} \cdot 2^x\right) \rfloor} = 2^{\lfloor x + \log\left(\frac{4}{3}\right) \rfloor} = 2^{\text{RDN}\left(x + \log\left(\frac{4}{3}\right) - \frac{1}{2}\right)} \approx 2^{\text{RDN}(x-0.084)} \,. \tag{19}$$

**Logarithmic unbiased quantization (LUQ)**   In the following, we suggest LUQ, a 4-bit unbiased estimation of neural gradients that apply stochastic pruning (Eq. (15)) to the 4-bit floating point quantizer $Q_\alpha(x)$

$$X_q = T_\alpha\left(Q_\alpha(x)\right) \,. \tag{20}$$

Since $T_\alpha$ and $Q_\alpha$ are unbiased, it follows by the law of total expectation that $X_q$ is an unbiased estimator for $x$:

$$E[X_q] = E\left[T_\alpha\left(Q_\alpha(x)\right)\right] = E\left[E\left[T_\alpha\left(Q_\alpha(x)\right)\right]|Q_\alpha(x)\right] = E\left[Q_\alpha(x)\right] = x \,, \tag{21}$$

where the expectation is over the randomness of $T_\alpha$ and $Q_\alpha$. In Fig. 2 we show an illustration of LUQ. The first step includes stochastic pruning for the values below the pruning threshold ($|x| < \alpha$). The second step includes the logarithmic quantization with format FP4 of the values above the pruning threshold ($|x| > \alpha$). In Fig. 3a we show an ablation study of the effect of the different parts of LUQ on ResNet50 ImageNet dataset - while standard FP4 diverges, adding stochastic-pruning or round-to-nearest-power allow to converge with significant degradation. Combining both methods improves the results and finally the suggested LUQ which includes additionally the suggested underflow threshold choice as the optimal unbiased value gets the best results.

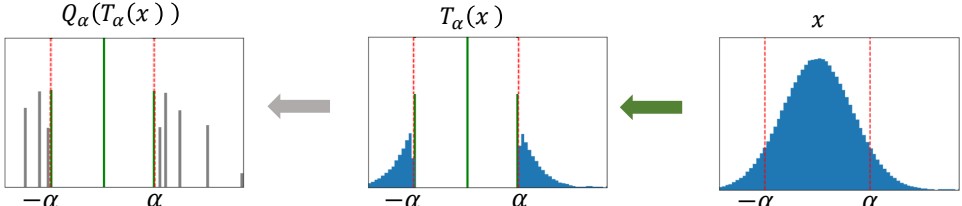

Figure 2: The effect of LUQ on the neural gradients histograms for one layer of ResNet18 Cifar100 dataset, with the underflow threshold $\alpha$ (red dashed line). The first step (green arrow) represents the effect of stochastic pruning (Eq. (15)) on the neural gradient. The second step (grey arrow) represents the logarithmic unbiased quantization (Eq. (16)), that quantize all the values $|x| > \alpha$.

### 3.1 SMP: REDUCING THE VARIANCE WHILE KEEPING IT UNBIASED

In the previous section, we presented an unbiased method for logarithmic quantization of the neural gradients called LUQ. Following the bias-variance decomposition, if the gradients are now unbiased, then the only remaining issue should be their variance. Therefore, we suggest a method to reduce the quantization variance by repeatedly sampling from the stochastic quantizers in LUQ, and averaging the resulting samples of the final weight gradients. The different samples can be calculated in parallel, so the only overhead on the network throughput will be the averaging operation. The power overhead will be $\sim \frac{1}{3}$ of the number of additional samples since it affects only in the update GEMM (Eq. (24)). For $N$ different samples, the proposed method will reduce the variance by a factor of $\frac{1}{N}$, without affecting the bias (Gilli et al., 2019). In Fig. 3b we show the effect of the different number of samples (SMP) on 2-bit quantization of ResNet18 Cifar100 dataset. There, we achieve with 16 samples accuracy similar to a full-precision network. This demonstrates that the variance is the only remaining issue in neural gradient quantization using LUQ, and that the proposed averaging method can erase this variance gap, with some overhead.

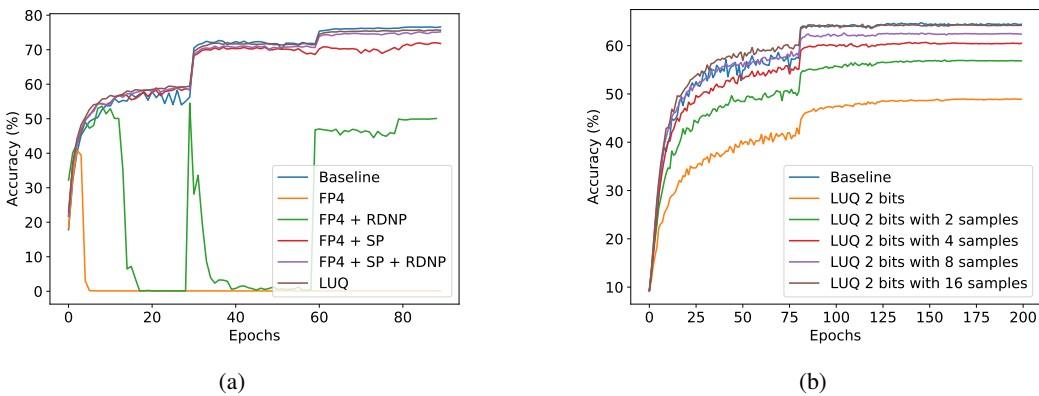

Figure 3: **(a):** ResNet50 top-1 validation accuracy in ImageNet dataset with different quantization schemes for the neural gradients. SP refers to stochastic pruning (Eq. (15)). RDNP refers to round-to-nearest-power (Eq. (19)). Notice that with the suggested LUQ we are able to almost close the degradation from baseline. **(b):** ResNet18 top-1 validation accuracy in CIFAR100 with quantization of the neural gradients to 2-bit (FP2 - [1,1,0] format) using different samples numbers to reduce the variance. Notice that 16 samples completely close the gap to the baseline.

### 3.2 FNT: FINE-TUNING IN HIGH PRECISION FOR ONE EPOCH

We suggest running one additional epoch in which we increase all the network parts to full-precision, except the weights which remain in low precision. We notice that with this scheme we get the best accuracy for the fine-tuned network. In inference time the activations and weights are quantized to lower precision. In Table 1 we can see the effect of the proposed fine-tuning scheme, improving the accuracy of the models by $\sim 0.4\%$.

## 4 EXPERIMENTS

In this section, we evaluate the proposed LUQ for 4-bit training on various DNN models. For all models, we use their default architecture, hyper-parameters, and optimizers combined with a custom-modified Pytorch framework that implemented all the low precision schemes. Additional experimental details appear in Appendix A.1.

**Main results**    In Table 1 we show the top-1 accuracy achieved in 4-bit training using LUQ to quantizing the neural gradients to FP4 and combined with a previously suggested method, SAWB (Choi et al., 2018a), to quantize the weights and activations to INT4. We compare our method with Ultra-low (Sun et al., 2020) showing better results in all the models, achieving SOTA in 4-bit training. Moreover, we improve the results with the two proposed schemes: neural gradients sampling (SMP - Section 3.1) and fine-tune in high precision (FNT - Section 3.2) achieving for the first time in 4-bit training 0.64% error in ResNet-50 ImageNet dataset, with our simple and hardware friendly methods. In Table 2 we apply the proposed LUQ on NLP models, achieving less than $0.4\%$ BLUE score degradation in Transformer-base model on the WMT En-De task.

**Overhead of SMP and FNT**    We limit our experiments with the proposed SMP method to only two samples. This is to achieve a similar computational overhead as Ultra-low, with their suggested two-phase-rounding (TPR) which also generates a duplication for the neural gradient quantization. The FNT method is limited to only 1 epoch to reduce the overhead in comparison to Ultra-low, which keeps the 1x1 convolutions in 8-bit. Specifically, the throughput of a 4-bit training network is 16x in comparison to high precision training (Sun et al., 2020). This means that doing one additional epoch in high precision reduces the throughput of ResNet-50 training by $\sim 16\%$ . In comparison, Ultra-low (Sun et al., 2020) does full-training with all the 1x1 convolutions in 8-bits, which reduces the throughput by $\sim 50\%$.

Table 1: Comparison of 4-bit training of the proposed method LUQ with Ultra-low (Sun et al., 2020) in various DNNs models with ImageNet dataset. FNT refers to fine-tune the trained model one additional epoch with the neural gradients at high precision (Section 3.2) and SMP refers to doing two samples of the SR quantization of neural gradients in order to reduce the variance (Section 3.1).

| Model | Baseline | Ultra-low | LUQ | LUQ + FNT | LUQ + SMP | LUQ + SMP + FNT |
|---|---|---|---|---|---|---|
| ResNet-18 | 69.7 % | 68.27% | 69.0% | 69.39 % | 69.1 % | 69.47 % |
| ResNet-50 | 76.5% | 74.01% | 75.32 % | 75.52 % | 75.63 % | 75.86 % |
| MobileNet-V2 | 71.9 % | 68.85 % | 69.69 % | 69.87 % | 69.9 % | 70.13 % |
| ResNext-50 | 77.6 % | N/A | 76.12 % | 76.39 % | 76.32 % | 76.55 % |

Table 2: Comparison of the BLUE score for 4-bit training of the proposed method LUQ with Ultra-low (Sun et al., 2020) in Transfomer base model on the WMT En-De task.

| Model | Baseline | Ultra-low | LUQ |
|---|---|---|---|
| Transfomer-base | 27.5 | 25.4 | 27.17 |

**Forward-backward ablations**    In Table 3 we show the top-1 accuracy in ResNet50 with different quantization schemes. The forward phase (activations + weights) is quantized to INT4 with SAWB (Choi et al., 2018a) and the backward phase (neural gradients) to FP4 with LUQ. As expected, the network is more sensitive to the quantization of the backward phase.

## 5 MF-BPROP: MULTIPLICATION FREE BACKPROPAGATION

The main problem of using different datatypes for the forward and backward phases is the need to cast them to a common data type before the multiplication during the backward and update phases. In our case, the weights ($W$) and pre-activations ($a$) are quantized to INT4 while the neural gradients ($\frac{\partial C}{\partial a}$) to FP4, where $C$ represent the loss function, $\phi$ a non-linear function and $v$ the post-activation. During the backward and update phases, in each layer $l$ there are two GEMMs between different

datatypes:

$$[\textbf{Forward}] \quad a_l = W_l v_{l-1} \qquad v_l = \phi(a_l) \tag{22}$$

$$[\textbf{Backward}] \quad \frac{\partial C}{\partial a_{l-1}} = \text{Diag}(\phi'(a_{l-1}) W_l^T \frac{\partial C}{\partial a_l} \tag{23}$$

$$[\textbf{Update}] \quad \frac{\partial C}{\partial W_l} = \frac{\partial C}{\partial a_l} a_l^T \tag{24}$$

Regularly, to calculate these GEMMs there is a need to cast both data types to a common data type (in our case, FP7 [1,4,2]), then do the GEMM and finally, the results are usually accumulated in a wide accumulator (Fig. 4a). This casting cost is not negligible. For example, casting INT4 to FP7 consumes $\sim 15\%$ of the area of an FP7 multiplier.

In our case, we are dealing with a special case where we do a GEMM between a number without mantissa (neural gradient) and a number without exponent (weights and activations), since INT4 is almost equivalent to FP4 with format [1,0,3]. We suggest transforming the standard GEMM block (Fig. 4a) to Multiplication Free BackPROP (MF-BPROP) block which contains only a transformation to standard FP7 format (see Fig. 4b) and a simple XOR operation. More details on this transformation appear in Appendix A.3. In our analysis (Appendix A.4) we show the MF-BPROP block reduces the area of the standard GEMM block by 5x. Since the FP32 accumulator is still the most expensive block when training with a few bits, we reduce the total area in our experiments by $\sim 8\%$. However, as previously showed (Wang et al., 2018) 16-bits accumulators work well with 8-bit training, so it is reasonable to think, it should work also with 4-bit training. In this case, the analysis (Appendix A.4) shows that the suggested MF-BPROP block reduces the total area by $\sim 22\%$.

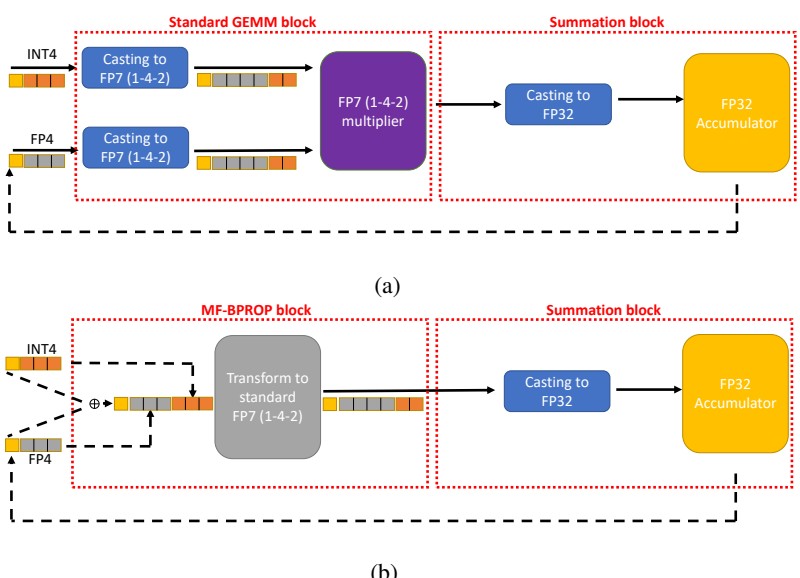

(a)

(b)

Figure 4: **(a):** Standard MAC block illustration containing the two main blocks - one for GEMM and second for accumulator. The GEMM block for hybrid datatype as in our case (FP4 and INT4) requires a casting to a common datatype before being inserted into the multiplier. **(b):** The suggested MAC block, which replace the multiplier with the proposed MF-BPROP. Instead of doing an expensive casting followed by a multiplication, we propose to make only a simple XOR and a transformation (Appendix A.3) reducing the GEMM area by 5x (Appendix A.4).

## 6 RELATED WORKS

Neural networks Quantization has been extensively investigated in the last few years. Most of the quantization research has focused on reducing the numerical precision of the weights and activations for inference (e.g., Courbariaux et al. (2016); Rastegari et al. (2016); Banner et al. (2019); Nahshan et al. (2019); Choi et al. (2018b); Bhalgat et al. (2020); Choi et al. (2018a); Liang et al. (2021)). In standard ImageNet models, the best performing methods can achieve quantization to 4 bits with small or no degradation Choi et al. (2018a). These methods can be used to reduce the computational resources in approximately a third of the training. However, without quantizing the neural gradients,

we cannot reduce the computational resources in the remaining two thirds of the training process. An orthogonal approach of quantization is low precision for the gradients of the weights in distributed training (Alistarh et al., 2016; Bernstein et al., 2018) in order to reduce the bandwidth and not the training computational resources.

Sakr & Shanbhag (2019) suggest a systematic approach to design a full training using fixed point quantization which includes mixed-precision quantization. Banner et al. (2018) first showed that it is possible to use INT8 quantization for the weights, activations, and neural gradients, thus reducing the computational footprint of most parts of the training process. Concurrently, Wang et al. (2018) was the first work to achieve full training in FP8 format. Additionally, they suggested a method to reduce the accumulator precision from 32bit to 16 bits, by using chunk-based accumulation and floating point stochastic rounding. Later, Wiedemann et al. (2020) showed full training in INT8 with improved convergence, by applying a stochastic quantization scheme to the neural gradients called non-subtractive-dithering (NSD) that induce sparsity followed by stochastic quantization. Also, Sun et al. (2019) presented a novel hybrid format for full training in FP8, while the weights and activations are quantized to [1,4,3] format, the neural gradients are quantized to [1,5,2] format to catch a wider dynamic range. Fournarakis & Nagel (2021) suggest a method to reduce the data traffic during the calculation of the quantization range.

While it appears that it is possible to quantize to 8-bits all computational elements in the training process, 4-bits quantization of the neural gradients is still challenging. Chmiel et al. (2021) suggested that this difficulty stems from the heavy-tailed distribution of the neural gradients, which can be approximated with a lognormal distribution. This distribution is more challenging to quantize in comparison to the normal distribution which is usually used to approximate the weights or activations (Banner et al., 2019).

Sun et al. (2020) was the first work that presented a method to reduce the numerical precision to 4-bits for the vast majority of the computations needed during DNNs training. They use known methods to quantize the forward phase to INT4 (SAWB (Choi et al., 2018a) for the weights and PACT (Choi et al., 2018b) for the activations) and suggested to quantize the neural gradients twice (one for the update and another for the next layer neural gradient) with a non-standard radix-4 FP4 format. The use of the radix-4, instead of the commonly used radix-2 format, allows covering a wider dynamic range. The main problem of their method is the specific hardware support for their suggested radix-4 datatype, which may limit the practicality of implementing their suggested data type.

Chen et al. (2020b) suggested reducing the variance in neural gradients quantization by dividing them into several blocks and quantizing each to INT4 separately. Their method requires each iteration to sort all the neural gradients and divide them into blocks, a costly operation that will affect the network throughput. Additionally, they suggested another method to quantize each sample separately. The multiple scales per layer in both methods do not allow the use of an efficient GEMM operation.

## 7 CONCLUSIONS

In this work, we analyze the difference between two rounding schemes: round-to-nearest and stochastic-rounding. We showed that, while the former has lower MSE and works better for the quantization of the forward phase (weights and activations), the latter is an unbiased approximation of the original data and works better for the quantization of the backward phase (specifically, the neural gradients).

Based on these conclusions, we propose a logarithmic unbiased quantizer (LUQ) to quantize the neural gradients to format FP4 [1,3,0]. Combined with a known method for quantizing the weights and activations to INT4 we achieved, without overhead, state-of-the-art in 4-bit training in all the models we examined, e.g., 1.18 % error in ResNet50 vs 2.49 % for the previous known SOTA (Sun et al. (2020)). Moreover, we suggest two more methods to improve the results, with overhead comparable to Sun et al. (2020). The first reduces the quantization variance, without affecting the unbiasedness of LUQ, by averaging several samples of stochastic neural gradients quantization. The second is a simple method for fine-tuning in high precision for one epoch. Combining all these methods, we were able for the first time to achieve 0.64 % error in 4-bit training of ResNet50 ImageNet dataset.

Finally, we exploit the special formats used for quantization (INT4 in the forward phase and FP4 format [1,3,0] in the backward phase) and suggest a block called MF-BPROP that avoids multiplication during two thirds of the training, thus reducing by 5x the area previously used by the multiplier.

## REPRODUCIBILITY

A source code of the experiments appears in the supplementary material. Additionally, in Appendix A.1 we added the details of the experiments, including the hyper-parameters used.

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

## A APPENDIX

### A.1 EXPERIMENTS DETAILS

In all our experiments we use the most common approach (Banner et al., 2018; Choi et al., 2018b) for quantization where a high precision of the weights are kept and quantized on-the-fly. The updates are done in full precision.

**ResNet / ResNext** We run the models ResNet-18, ResNet-50 and ResNext-50 from torchvision. We use the standard pre-processing of ImageNet ILSVRC2012 dataset. We train for 90 epochs, use an initial learning rate of 0.1 with a 0.1 decay at epochs 30,60,80. We use standard SGD with momentum of 0.9 and weight decay of 1e-4. The minibatch size used is 256. Following the DNNs quantization conventions (Banner et al., 2018; Nahshan et al., 2019; Choi et al., 2018b) we kept the first and last layer (FC) at higher precision. Additionally, similar to Sun et al. (2020) we adopt the full precision at the shortcut which constitutes only a small amount of the computations ($\sim 1\%$). The "underflow threshold" in LUQ is updated in every bwd pass as part of the quantization of the neural gradients. In all experiments, the BN is calculated in high-precision.

**MobileNet V2** We run Mobilenet V2 model from torchvision. We use the standard pre-processing of ImageNet ILSVRC2012 dataset. We train for 150 epochs, use an initial learning rate of 0.05 with a cosine learning scheduler. We use standard SGD with momentum of 0.9 and weight decay of 4e-5. The minibatch size used is 256. Following the DNNs quantization conventions (Banner et al., 2018; Nahshan et al., 2019; Choi et al., 2018b) we kept the first and last layer (FC) at higher precision. Additionally, similar to Sun et al. (2020) we adopt the full precision at the depthwise layer which constitutes only a small amount of the computations ($\sim 3\%$). The "underflow threshold" in LUQ is updated in every bwd pass as part of the quantization of the neural gradients. In all experiments, the BN is calculated in high-precision.

**Transformer**    We run the Transformer-base model based on the Fairseq implementation on the WMT 14 En-De translation task. We use the standard hyperparameters of Fairseq including Adam optimizer. We implement LUQ over all attention and feed forwards layers.

Table 3: ResNet-50 accuracy with ImageNet dataset while quantization different parts of the network. The forward phase is quantized to INT4 format with SAWB (Choi et al., 2018a) while the backward phase is quantized with the proposed LUQ. As expected, the quantization of the backward phase makes more degradation to the network accuracy.

| Forward | Backward | Accuracy |
|---------|----------|----------|
| FP32 | FP32 | 76.5 % |
| INT4 | FP32 | 76.35 % |
| FP32 | FP4 | 75.6 % |
| INT4 | FP4 | 75.32 % |

### A.2    ADDITIONAL EXPERIMENTS

LUQ requires the measurement of the maximum of the neural gradient in order to get the underflow threshold (Section 3). In Fig. 5a we compare the proposed LUQ with the method proposed in Fournarakis & Nagel (2021) which suggests reducing the data movement overhead that occurs in the calculation on-the-fly of the quantization ranges by using a running average of the previous iterations statistics. As we can notice, the limited dynamic range in 4-bit quantization requires an exact statistics measurement of the tensor since the proposed approximation induces significant accuracy degradation. The SMP method (Section 3.1) has a power overhead of $\sim \frac{1}{3}$ of the number of additional samples since it influences only the update GEMM. In Fig. 5b we compare LUQ with one additional sample which has $\sim 33\%$ power overhead with regular LUQ with additional $\sim 33\%$ epochs. The lr scheduler was expanded respectively. We can notice that even both methods have similar overhead the variance reduction achieved with SMP is more important for the network accuracy than increasing the training time.

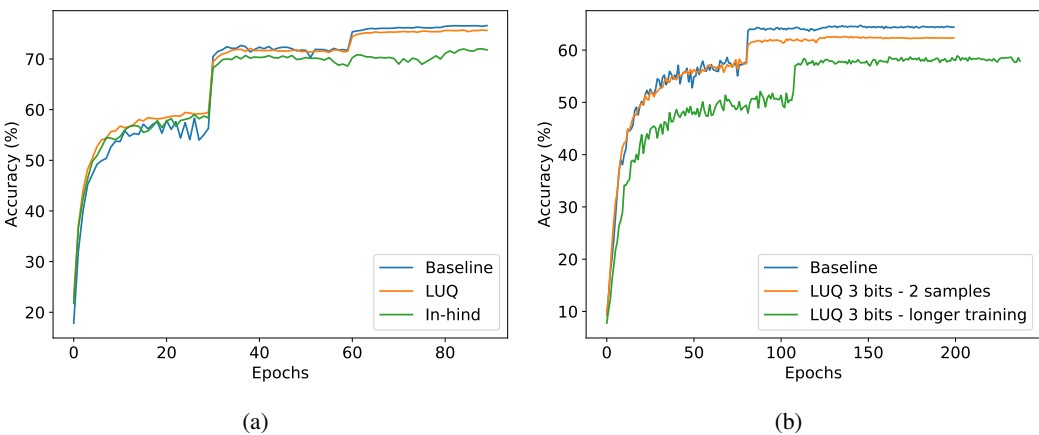

(a)                                                    (b)

Figure 5: **(a):** Comparison of the top-1 validation accuracy in ResNet-50 ImageNet dataset with the proposed LUQ and with the quantization dynamic range approximation in In-hind Fournarakis & Nagel (2021) in 4-bit training. **(b):** Comparison of ResNet-18 3 bit training on Cifar100 dataset of LUQ with 2 samples with longer training of regular LUQ. Both methods have similar overhead, but the SMP method leads to better accuracy.

### A.3    TRANSFORM TO STANDARD FP7

We suggest a method to avoid the use of an expensive GEMM block between the INT4 (activation or weights) and FP4 (neural gradient). It includes 2 main elements: The first is a simple xor operation between the sign of the two numbers and the second is a transform block to standard FP7 format. In

Fig. 6 we present an illustration of the proposed method. The transformation can be explained with a simple example: for simplicity, we avoid the sign which requires only xor operation. The input arguments are 3 (011 bits representation in INT4 format) and 4 (011 bits representation in FP4 1-3-0 format). The concatenation brings to the bits 011 011. Then looking at the table in the input column where the M=3 (since the INT4 argument = 3) and get the results in FP7 format of 0100 10 ( = E+1 2) which is 12 in FP7 (1-4-2) as the expected multiplication result.

In the next section, we analyze the area of the suggested block in comparison to the standard GEMM block, showing a 5x area reduction.

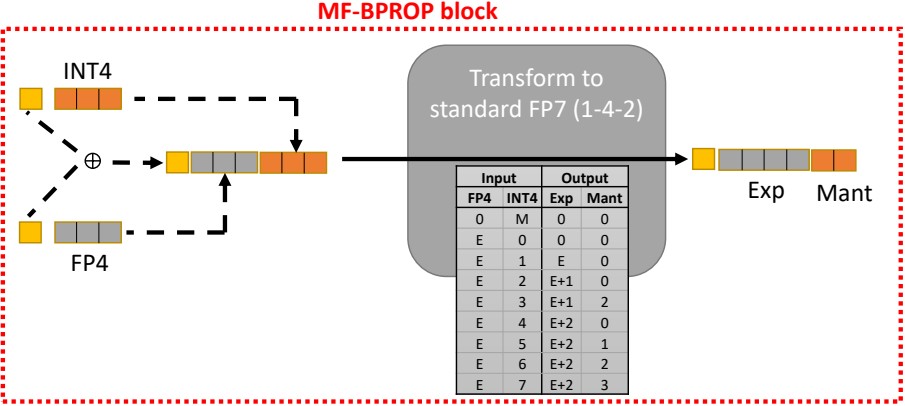

Figure 6: Illustration of MF-BPROP block which replaces a standard multiplication. It includes: (1) a simple xor operation between the sign. (2) A transform to standard FP7 format. We present the table to make this transform - E and M represent the bits of the FP4 and INT4 respectively without the sign. Exp and Mant are the bits of the output exponent (4-bit) and mantissa (2-bit) of the output in FP7 format.

## A.4 Backpropagation without multiplcation analysis

In this section, we show a rough estimation of the logical area of the proposed MF-BPROP block which avoids multiplication and compares it with the standard multiplier. In hardware design, the logical area can be a good proxy for power consumption (Iman & Pedram, 1997). Our estimation doesn't include synthesis optimization. In Table 4 we show the estimation of the number of gates of a standard multiplier, getting 264 logical gates while the proposed MF-BPROP block has an estimation of 49 gates (Table 5) achieving a $\sim 5x$ area reduction. For fair comparison we remark that in the proposed scheme the FP32 accumulator is the most expensive block with an estimation of 2453 gates, however we believe it can be reduced to a narrow accumulator such as FP16 (As previously shown in Wang et al. (2018) which have an estimated area of 731 gates. In that case, we reduce the total are by $\sim 22\%$.

Table 4: Rough estimation of the number of logical gates for a standard GEMM block which contain two blocks: a casting to FP7 and a FP7 multiplier.

| Block | Operation | # Gates |
|---|---|---|
| Casting to FP7 | Exponent 3:1 mux | 12 |
| | Mantissa 4:1 mux | 18 |
| FP7 [1,4,2] multiplier | Mantissa multiplier | 99 |
| | Exponent adder | 37 |
| | Sign xor | 1 |
| | Mantissa normalization | 48 |
| | Rounding adder | 12 |
| | Fix exponent | 37 |
| **Total** | | **264** |

Table 5: Rough estimation of the number of logical gates for the proposed MF-BPROP block.

| Block | Operation | # Gates |
|---|---|---|
| MF-BPROP | Exponent adder | 30 |
| | Mantissa 4:1 mux | 18 |
| | Sign xor | 1 |
| **Total** | | **49** |

