# OpenReview forum: "Logarithmic Unbiased Quantization: Practical 4-bit Training in Deep Learning"
_ICLR.cc/2022/Conference — ICLR 2022 Submitted_

### Official Review · Reviewer_Sxpj · 2021-10-30

**Correctness:** 3
**Technical Novelty And Significance:** 3
**Empirical Novelty And Significance:** 3
**Recommendation:** 6
**Confidence:** 4

**Main Review:**

** Strength **
1. Overall the paper is well written, quite thorough, and well-cited. The theoretical analysis related to stochastic rounding and rounding-to-nearest are well justified.
2. The paper achieved superior training results in 4-bit training.
3. The authors not only provided the 4-bit training algorithm but also suggested dedicated hardware blocks.
** Weaknesses **
1. Some parts of the proposed method still need high-precision
2. Some experimental results, MobileNet-V2 and ResNext-50, are not provided because of the time deadline.
3. The hardware implementation costs are evaluated in terms of logical area, but I think it should also be analyzed in terms of memory area since the multiple sampling (SMP) and underflow threshold computation parts need additional memories.

** minor comments **
1. In Figure 1 (b) and (c), which rounding methods are adopted for Bwd and Fwd, respectively.
2. In order to reduce variance, the author performs multiple samplings and averages them, but as the sampling increases, does not it get closer to the RDN?
3. There is a typo on page 5. (The a different samples -> The different samples)
4. The authors mentioned that "we increase all network parts to full-precision, except the weights" for the high-precision fine-tuning. If activation is also trained with full precision here, after fine-tuning, the activation precision should be lowered back to 4 bits.  Was there any performance degradation at this point?


**Summary Of The Paper:**

The paper proposes some techniques to train a deep neural network in 4-bit and provides a theoretical analysis. The experimental results support the effectiveness of the proposed method. Although some part of the method still needs high precision, the Reviewer thinks that it could open a door to training the deep neural network with ultra-low bits.

**Summary Of The Review:**

This paper proposed several techniques for effective 4-bit training. Some parts of the proposed method still require high-precision, but it might be a good starting point for full 4-bit training in the near future.

---

> ### Author Response · Authors · 2021-11-15
> **Answer to reviewer Sxpj**
>
> $\textbf{Q1:}$ "Some parts of the proposed method still need high-precision".
>
> $\textbf{A1:}$ The proposed method LUQ to quantize the neural gradients to FP4 in general does not require high-precision.
> If the reviewer refers to the suggested method FNT, which is an extension for improving the results, please note that in all our experiments we limit ourselves to only 1 epoch of FNT which means only 16 \% overhead, which is smaller than the overhead of the previous method [1] (see "Overhead of SMP and FNT" in section 4). We show the results without the FNT and we still are SOTA in 4-bit training.
>
>
> $\textbf{Q2:}$ "Some experimental results, MobileNet-V2 and ResNext-50, are not provided because of the time deadline."
>
> $\textbf{A2:}$ We updated the MobileNet-V2 and ResNext-50 results in the new revision of the paper. Additionally, we added an experiment with a Transformer-base model.
>
> $\textbf{Q3:}$ "The hardware implementation costs are evaluated in terms of logical area, but I think it should also be analyzed in terms of memory area since the multiple sampling (SMP) and underflow threshold computation parts need additional memories."
>
> $\textbf{A3:}$ The suggested method LUQ to quantize the neural gradient to FP4 is orthogonal to the hardware implementation.
> In LUQ we suggest a method to quantize the neural gradients to FP4 format 1-3-0 (sign-exponent-mantissa) without bias. The calculation of the threshold doesn't need additional memory or computation in comparison to INT quantization or the layer-wise gradient scaling used in [1] gradient since both required an all-reduce operation. We agree with the reviewer that the multiple-sampling requires extra computation (not memory) in comparison to standard training but it is completely computationally equivalent to the previous method TPR for 4-bit training used in [1].
> The hardware implementation suggestion called MF-BPROP is a general hardware implementation that can avoid multiplications when one argument has only mantissa (like INT4 in our case) and the other argument only exponent (like FP4 1-3-0 in our case). We do not limit our method to any kind of quantization. For this general hardware implementation, the important measure is the logical area it takes in comparison to standard multiplier.
>
> $\textbf{Q4:}$ "In Figure 1 (b) and (c), which rounding methods are adopted for Bwd and Fwd, respectively."
>
> $\textbf{A4:}$ In figure 1(b) and (c) the bwd and fwd passes, are respectively in full-precision to focus only on the effect of the rounding scheme in one part (Fwd or Bwd) in each experiment. We clarify it in the caption of Fig.1 in the new revision.
>
> $\textbf{Q5:}$ "In order to reduce variance, the author performs multiple samplings and averages them, but as the sampling increases, does not it get closer to the RDN?"
>
> $\textbf{A5:}$ As the number of samples goes to infinity we get closer to the real-value gradients. \\
>
>
> $\textbf{Q6:}$ "The authors mentioned that "we increase all network parts to full-precision, except the weights" for the high-precision fine-tuning. If activation is also trained with full precision here, after fine-tuning, the activation precision should be lowered back to 4 bits. Was there any performance degradation at this point?"
>
> $\textbf{A6:}$ The reviewer comment is right, the activations are trained during the fine-tuning in high precision. In the inference time, they are lowered back to 4 bits. We notice empirically that this scheme improves the results than doing the fine-tuning in 4 bits for the activations. We added a better explanation of it in Sec 3.2. Notice that as remarked in the experiments section, in all the experiments the fine-tune is done only for 1 additional epoch, to reduce the overhead and be comparable with previous methods overhead.
>
> -----------------------------------------------------------------------------------------------------------------------------------------------------------------------------
>
> [1]  Xiao Sun, Naigang Wang, Chia-Yu Chen, Jiamin Ni, A. Agrawal, Xiaodong Cui, Swagath Venkatara-mani, K. E. Maghraoui, V. Srinivasan, and K. Gopalakrishnan. Ultra-low precision 4-bit training of deep neural networks. In NeurIPS, 2020

---

> > ### Comment · Area_Chair_BECa · 2021-12-01
> > **any further thoughts?**
> >
> > Reviewer Sxpj, thanks for the thoughtful review! Any further thoughts after reading the authors' response?

---

### Official Review · Reviewer_t7UD · 2021-11-01

**Correctness:** 3
**Technical Novelty And Significance:** 3
**Empirical Novelty And Significance:** 2
**Recommendation:** 5
**Confidence:** 4

**Details Of Ethics Concerns:**

Not applicable.

**Main Review:**

While this paper is interesting and presents promising results, I did have many questions listed hereafter:

Introduction: Radix-4 representation is criticized arguing that conversion to radix-2 requires an explicit multiplication to modify exponent and mantissa. This is untrue, in the paper by Sun et al. that the authors cite, the radix-4 format does not use a mantissa, it only uses 3 exponent bits, and therefore converting to radix-2 simply requires appending the exponent field by a zero.

Section 2: In the calculation of rounding variance and MSE, why is the input x considered deterministic?

In Section 2, conclusion, the following claim is made:  "the forward phase should be quantized deterministically (using RDN) since stochastic rounding will not make help making the loss estimate unbiased (due to the non-linearity of the loss and activation functions) while unnecessarily increasing the mean-square-error". Here the authors are claiming that the use of non-linear activations and loss will negate the fact that SR is unbiased. Why is that the case? Is this a known result from prior arts? If so, the authors should add a reference as was done in this same paragraph regarding the works of Chen et al., and Bottou.

The LUQ proposed in eq. (11)-(12) is not unbiased when n=b-1, i.e., in the quantization region corresponding to the largest magnitudes. Usually, such boundary cases do not matter, but given that here there are only 16 regions, and that the data is assumed to be heavy-tailed (most of the data would actually fall in this final region), this issue might be significant. Can the authors comment on what happens in the final quantization region?

Related to the above, this is more of a suggestion. Rather than setting an underflow value as quantization meta-parameter, why isn't the max of the tensor used instead? This would avoid the above issue. Have the authors considered that? Did tuning an underflow hyperparameter yield better results?

In eq.(14), why is a subtraction of a 1/2 term needed when RDN is defined as per eq.(5)? Also this equation is improperly typeset.

One final question: The issue of quantization bias occurring in gradients and weight updates has previously been studied in the paper below. Can the authors compare their findings and check if their conclusions are consistent with prior arts?
Sakr et al., Per-Tensor Fixed-Point Quantization of the Back-Propagation Algorithm, in ICLR 2019.

The experimental results, and crucially, the implementation details in the Appendix look good.

Finally, there are many typos and grammatical errors throughout the paper. I urge the authors to perform a spell check.

**Summary Of The Paper:**

This paper proposes techniques to quantize the gradients in the back-propagation algorithm down to 4 bits. To do so, the authors propose the [1,3,0] radix-2 FP4 format, as well as bias removal and variance reduction techniques. In order to achieve SOTA accuracy, the method requires an additional stage of high precision fine-tuning.

**Summary Of The Review:**

The paper is interesting, tackles an important problem, and presents promising results. Therefore, I do not recommend a clear reject. However, there are unfortunately many issues I though were unclear and hence raised in my detailed review. Therefore, I cannot recommend acceptance of the paper if these issues are not addressed. Hence, for now I am rating the paper as a weak reject.

---

> ### Author Response · Authors · 2021-11-15
> **Answer to reviewer t7UD - Part 1**
>
> $\textbf{Q1:}$ "Radix-4 representation is criticized arguing that conversion to radix-2 requires an explicit multiplication to modify exponent and mantissa. This is untrue, in the paper by Sun et al. that the authors cite, the radix-4 format does not use a mantissa, it only uses 3 exponent bits, and therefore converting to radix-2 simply requires appending the exponent field by a zero."
>
> $\textbf{A1:}$ In order to convert an fp32 number in the conventional radix-2 to a radix-4 number it requires an explicit multiplication with the number 1.6 as explained in Figure 2(c) in the paper by Sun et al [1]. If we understand correctly the reviewer suggestion is to take a number quantized in radix-2 and by multiplying it by 2 (append zero in the exponent) we get radix-4. We can show with a simple example that this is not true: let's assume radix-2 quantization with bins 1,2,4,8 and radix-4 quantizations with bins 1,4,16,64. For the number 4.5, if we quantize it first to radix-2 we get the quantized number 4, then we multiply it by 2 we get 8. In contrast, radix-4 quantization should give the result 4.
>
>
> $\textbf{Q2:}$ "Section 2: In the calculation of rounding variance and MSE, why is the input x considered deterministic?."
>
> $\textbf{A2:}$ Section 2 analysis is an analysis of the MSE of a given input x under different rounding schemes, meaning that x is deterministic. The stochasticity we average on is only due to the noise of the quantizer (when it exists).
>
>
> $\textbf{Q3:}$ "[...] Here the authors are claiming that the use of non-linear activations and loss will negate the fact that SR is unbiased. Why is that the case? Is this a known result from prior arts? If so, the authors should add a reference as was done in this same paragraph regarding the works of Chen et al., and Bottou"
>
> $\textbf{A3:}$ In order the make our claim more clear, we added section 2.1 to the new revision. In this section, we showed that the linearly of the backpropagation induces that unbiasedness at the tensor level causes unbiasedness at the model level. However, in the forward pass, the non-linear activations break the unbiasedness so even if we get unbiasedness at the tensor level it does not induce at the model level.
>
>
> $\textbf{Q4:}$ " The LUQ proposed in eq. (11)-(12) is not unbiased when n=b-1, i.e., in the quantization region corresponding to the largest magnitudes. Usually, such boundary cases do not matter, but given that here there are only 16 regions, and that the data is assumed to be heavy-tailed (most of the data would actually fall in this final region), this issue might be significant. Can the authors comment on what happens in the final quantization region?"
>
> $\textbf{A4:}$ In general case the reviewer is right, for any value that is bigger than the maximum bin we will create a bias. However, in LUQ we choose the maximum of the tensor to be equal to the maximum bin (see "underflow threshold" paragraph in Sec.3). With this (critical) choice we avoid creating any bias in the heavy-tail region and create a completely unbiased logarithmic quantization.
>
> $\textbf{Q5:}$ " Rather than setting an underflow value as quantization meta-parameter, why isn't the max of the tensor used instead? This would avoid the above issue. Have the authors considered that? Did tuning an underflow hyperparameter yield better results? "
>
> $\textbf{A5:}$ The reviewer is right, as mentioned in A4 above we chose the underflow threshold to be the optimal unbiased value. In other words, the maximum of the tensor is equal to the maximum quantization bin. Using this threshold we were able to get a fully unbiased logarithmic quantization and get the best results in comparison to other thresholds that we tried.
>
>
> $\textbf{Q6:}$ "In eq.(14), why is a subtraction of a 1/2 term needed when RDN is defined as per eq.(5)? Also this equation is improperly typeset."
>
> $\textbf{A6:}$ In equation 19 (previously equation 14) we show the implementation of the round-to-nearest-power (RDNP). We show that it can be done only by moving the RDN operation to the exponent and adding a constant (~ -0.084). The 1/2 term is required to transform a floor operation, as it is in the third element of equation 19 (previously equation 14), to round-to-nearest (i.e., floor(x) = rdn(x-0.5) ). Eq. 5 is the bias that RDN operation induces. Typesetting fixed, thanks.

---

> > ### Comment · Reviewer_t7UD · 2021-11-25
> > **Thanks for the response**
> >
> > Q1: I think it is important to clarify if we want to convert from Radix-4 to Radix-2 or vice versa. The former is trivial when it comes to exponent handling. If the QAT keeps data in Radix-4 representation, then using this data in a Radix-2 datapath should be straightforward.
> >
> > Q2: Ok, but why were input statistics not considered. This is a limitation.
> >
> > Q3: New explanation is appreciated. I don't think it is fit to compared biased noise to dropout - the two are very different. Also, I do not believe the use of non-linear activation makes it OK to use biased noise. Perhaps one could argue that negating this noise can be achieved via the learned bias parameter. In any case, all these hand-wavy explanations do reinforce that this was not a known result.
> >
> > Q4, Q5 & Q6: OK, perhaps the authors should make this clearer in the manuscript.
> >
> > Q7: OK.
> >
> > Overall, I appreciate the responses. However, as mentioned above, not all concerns were addressed. I am therefore maintaining my original score of 5.

---

> > > ### Author Response · Authors · 2021-11-26
> > > **Answer to the response of reviewer t7UD**
> > >
> > >  $\textbf{Q1:}$ "I think it is important to clarify if we want to convert from Radix-4 to Radix-2 or vice versa. The former is trivial when it comes to exponent handling. If the QAT keeps data in Radix-4 representation, then using this data in a Radix-2 datapath should be straightforward."
> > >
> > >
> > >  $\textbf{A1:}$ We agree with the reviewer that to transform a number that is quantized in Radix-4 to Radix-2 is trivial. However, it is not the case in Ultra-low [1], since the accumulators used there have a radix-2 representation (usually fp32).
> > >  Specifically, the backprop process in [1] requires, before the GEMM operation, the result of the accumulator of the previous layer, which is represented in fp32 radix-2, would be quantized to fp4 radix-4 representation. This quantization requires an explicit multiplication, as explained also in Figure 2(c) in [1].
> > >
> > >  $\textbf{Q2:}$ Ok, but why were input statistics not considered. This is a limitation.
> > >
> > >  $\textbf{A2:}$ Apologies if this was not clear: since our results hold for any deterministic (fixed) input x, they hold for any probability distribution on the input. For example, our main point in section 2 was eq. 9:
> > >  $\[\forall x: \mathrm{MSE}(\mathrm{SR}(x)) \geq \mathrm{MSE}(\mathrm{RDN}(x))\]$
> > >  Now, suppose $x$ follows a probability distribution $p(x)$, then we can take expectation over $x$ with respect to this distribution
> > >   $\mathbb{E}_{x\sim p(x)} [ \mathrm{MSE}(\mathrm{SR}(x))]  \geq $
> > >
> > > $\mathbb{E}_{x\sim p(x)} [ \mathrm{MSE}(\mathrm{RDN}(x))]$.
> > >  So the MSE of SR is always greater or equal than the MSE of RDN, for any $p(x)$.
> > >  Since we do not need to make any assumption on the input statistics, we believe that the fact that our analysis is true for $\textbf{any}$ $x$ is not a limitation, but an advantage.
> > >
> > >
> > >  $\textbf{Q3.a:}$: [...] I do not believe the use of non-linear activation makes it OK to use biased noise. Perhaps one could argue that negating this noise can be achieved via the learned bias parameter. In any case, all these hand-wavy explanations do reinforce that this was not a known result.
> > >
> > >  $\textbf{A3.a:}$  Apologies if we were not clear, but we did not say that the non-linear activations make it ok to use biased noise---we said that the non-linearity of the activations (and the loss) anyway creates a bias that cannot be fixed with an unbiased quantization. In other words, when using an unbiased quantization in the forward path, the non-linear function will create a bias in the next layer. For example, suppose we have two layers with weights $W_1, W_2$, activation $\sigma$, input $x$, the SR quantizer $Q$, and denote as $\mathbb{E}$ the expectation with respect to the randomness in $Q$. Then, despite that $Q$ is unbiased (i.e., $\mathbb{E}[Q(x)]=x$), we get
> > >  $\[\mathbb{E}[ \sigma (W_2 Q (\sigma ( W_1 x))]  \neq  \mathbb{E}[ \sigma (W_2  (\sigma ( W_1 x))] \]$
> > >  since $\sigma$ is non-linear.
> > >  This means there is no point to use SR in the forward path, since it will increase the MSE (from eq. 9), but it will not fix the bias issue. In contrast, the backward pass (as represented in eq. 13) is linear, so the unbiased quantization of the neural gradient (LUQ) will create an unbiased update (as shown in eq. 14).  The conclusion of this analysis is that the forward path it is not recommended to use SR but SR is recommended in the backward path.
> > >
> > >  $\textbf{Q3.b:}$:  I don't think it is fit to compared biased noise to dropout - the two are very different.
> > >
> > >  $\textbf{A3.b:}$ Apologies if we were not clear, but we did not mean to make such a comparison. We added the mention to dropout following a (good) comment by reviewer 9cKC that, similarly to SR in the forward pass, dropout is unbiased (on its input) and increases the MSE, but, in contrast to SR in the forward pass, dropout improves generalization. So, we acknowledged that increasing the MSE might improve generalization in some cases, though such increase in MSE is typically considered a bad thing in quantization papers.
> > >
> > >
> > >  $\textbf{Q4:}$, "Q4, Q5 , Q6: OK, perhaps the authors should make this clearer in the manuscript."
> > >
> > >  $\textbf{A4:}$ We will clarify these points, as well as Q1,Q2,Q3, in the final version of the manuscript.
> > >
> > >  Many thanks for your help in improving this manuscript. Please let us know if there are any remaining concerns. If not, we kindly ask the reviewer to increase the score.

---

> > > > ### Comment · Area_Chair_BECa · 2021-12-01
> > > > **any additional comments?**
> > > >
> > > > Thanks authors and reviewer for this discussion! To the reviewer, any final thoughts in response to the authors' follow-up here?

---

> > > > > ### Comment · Reviewer_t7UD · 2021-12-01
> > > > > **Thanks to the authors for the follow-up**
> > > > >
> > > > > Thank you to the authors for taking the time to follow-up on my earlier comments. At this point, I no longer have any questions and maintain my current score of marginal reject as I believe this paper is very interesting but has a significant amount of shortcomings that need to be addressed before publication.

---

> ### Author Response · Authors · 2021-11-15
> **Answer to reviewer t7UD - Part 2**
>
> $\textbf{Q7:}$ "The issue of quantization bias occurring in gradients and weight updates has previously been studied in the paper below. Can the authors compare their findings and check if their conclusions are consistent with prior arts? Sakr et al., Per-Tensor Fixed-Point Quantization of the Back-Propagation Algorithm, in ICLR 2019.."
>
> $\textbf{A7:}$ We thank the reviewer for this reference, we added it to the previous works papers in the new revision. This paper suggests a systematic approach, based on five criterions, to design full training using fixed point quantization. One of the criterions is to reduce the bias of the weight gradients. This is completely consistent with our analysis - as mentioned in the "conclusion" paragraph in sec.2: "It was previously proved that unbiased neural gradients quantization leads to an unbiased
> estimate of the weight gradients (e.g., Chen et al. (2020a) [3])". In our work, LUQ, we focus on an unbiased quantization of the neural gradients, in order to create an unbiased estimation of the weight gradients.
>
> -----------------------------------------------------------------------------------------------------------------------------------------------------------
>
> [1]  Xiao Sun, Naigang Wang, Chia-Yu Chen, Jiamin Ni, A. Agrawal, Xiaodong Cui, Swagath Venkatara-mani, K. E. Maghraoui, V. Srinivasan, and K. Gopalakrishnan. Ultra-low precision 4-bit training of deep neural networks. In NeurIPS, 2020
> [3] Jianfei Chen, Yu Gai, Zhewei Yao, Michael W Mahoney, and Joseph E Gonzalez. A statistical
> framework for low-bitwidth training of deep neural networks. arXiv preprint arXiv:2010.14298,
> 2020a.

---

### Official Review · Reviewer_na7b · 2021-11-02

**Correctness:** 3
**Technical Novelty And Significance:** 3
**Empirical Novelty And Significance:** 3
**Recommendation:** 6
**Confidence:** 4

**Main Review:**


Strong points:
* Theoretically well-motivated choice of quantization scheme for forward and backward path.
* The logarithmic unbiased quantization method is theoretically sound and well explained.
* Strong results for 4 bit training of various ImageNet models.
* Clear ablation studies for the contribution of SMP and FNT in table 1.
* Most parts of the paper are well written and easy to follow.

Weak points:
* It is unclear what the contributions of unbiased stochastic pruning and logarithmic unbiased rounding in LUC are. An ablation study would be insightful on this.
* The SMP method for reducing variance requires further clarification. It is unclear where and when this exactly is applied. For example, are the multiple samples only used to calculate the weight update (similar to the second sample in Sun et al.) or is the average of multiple samples also used in back propagating to he other layers. If the latter, then this seems to likely imply that a higher bit width needs to be used in the matmul of the next layer. How does the multiple samples compare in performance and complexity to using a different bit width?
* FNT: full-precision fine-tuning makes the comparison to other low-bit training methods unfair. But results are also show without it and show competitive results.
* The transform to Standard FP7 lacks detail and explanation. It is hard to understand how the input/output table is constructed.
* While the paper is easy to read and clearly written, many small details are missing, especially on the experimentation part (see questions below).

Questions:
* What is the HW impact of rounding to nearest power? This seems significantly more complicated than adding uniform noise as it is the case for uniform quantization.
* How do they define the pruning threshold? Are they learned/updated? I could not find any details on this in the paper.
* How do you deal with BN?
* What quantization approach is applied in the forward pass (weights and activations)? Rounding to nearest is clear, but how are the ranges defined/learned etc.

Editorial notes:
* Few appreciations introduced are unclear what they stand for. E.g SMP, FNT, RDN. The later I assume comes from rounding-to-nearest, but would than not RTN be the right appreciation?
* Typo below equation 9 (rounding-to-zero).
* Page 3: “will not make help making the loss estimate unbiased”

**Summary Of The Paper:**

The paper focuses on reducing the computational footprint of training neural networks using quantization. They investigate the importance of having unbiased gradients and show that stochastic rounding is important for the backwards pass. They propose a logarithmic unbiased quantization (LUQ) scheme which achieves SOTA results for 4-bit training on ImageNet. Results are further improved with multiple sampling (SMP) and a final full precision fine-tuning (FNT) at the cost of extra compute overhead. Finally they introduce a HW block which exploits LUQ by avoiding multiplication which can reduce the multiplier area by 5 times.

**Summary Of The Review:**

Overall a well written paper that is easy to follow (except a few parts highlighted above). The proposed approach is somewhat novel, it combines stochastic rounding with logarithmic quantization and stochastic pruning. The main strength are that most parts are well motivated and the strong 4 bit training results. The main weaknesses are around SMP and some other parts that are unclear. Overall the contributions slightly outweigh the weaknesses of the paper.

---

> ### Author Response · Authors · 2021-11-15
> **Answer to reviewer na7b - Part 1**
>
> $\textbf{Q1:}$ "It is unclear what the contributions of unbiased stochastic pruning and logarithmic unbiased rounding in LUC are. An ablation study would be insightful on this. "
>
> $\textbf{A1:}$ Perhaps this was missed, but in Fig 3(a) we showed such an ablation study for ResNet-50 ImageNet dataset. Standard FP4 diverges, while adding SP (Eq 10) is able to converge with some degradation. Adding to it the round-to-nearest-power (RDNP) improves even more the results. Finally, LUQ, which includes all the methods (as explained in section 3) gets the best results.
>
> $\textbf{Q2:}$ " The SMP method for reducing variance requires further clarification. It is unclear where and when this exactly is applied. For example, are the multiple samples only used to calculate the weight update (similar to the second sample in Sun et al.) or is the average of multiple samples also used in back propagating to he other layers. If the latter, then this seems to likely imply that a higher bit width needs to be used in the matmul of the next layer. How does the multiple samples compare in performance and complexity to using a different bit width?"
>
> $\textbf{A2:}$ The SMP method is only applied to calculate the weight update. Moreover, in order to compares the overhead of this method, and following reviewer 9cKC suggestion, we added in Fig 5(b) an experiment that compares the SMP method with a longer training regime to get comparable overhead. In that case, SMP has signficantly better accuracy. Lastly, we agree with the reviewer that a different approach can be increasing the bitwidth on part of the network as was done in [1] but it's outside of the scope of our work, since we focus on 4-bit training. Notice that such approach requires an analysis in which part of the network we should increase the bitwidth, the analysis can vary between different models and datasets.
>
> $\textbf{Q3:}$ " FNT: full-precision fine-tuning makes the comparison to other low-bit training methods unfair. But results are also show without it and show competitive results."
>
> $\textbf{A3:}$ In general we agree with the reviewer that FNT can make the comparison unfair. But, as mentioned in the experiments section we limit our experiments to only 1 epoch of FNT getting an overhead of $\sim 16 \%$. This overhead is still lower than previous methods [1] that suggest running the 1x1 convolutions in fp8 getting and overhead of $\sim 50 \%$ in ResNet50.
>
> $\textbf{Q4:}$ " The transform to Standard FP7 lacks detail and explanation. It is hard to understand how the input/output table is constructed."
>
> $\textbf{A4:}$ MF-BPROP can avoid multiplication when one of the arguments has only a mantissa (like INT4) and the other argument only has an exponent (like our FP4). For simplicity, let's explain the transform without the sign which requires only xor operation. The input arguments are 3 (011 bits representation in INT4 format) and 4 (011 bits representation in FP4 1-3-0 format). The concatenation results in 011 011. Then we look at the table at input where the M=3 (since our INT4 argument = 3) and get the results in FP7 format of 0100 10 ( = E+1 2) which is 12 in FP7 (1-4-2) as the expected multiplication result. We added this example in section A.4 in the new revision.
>
> $\textbf{Q5:}$ "What is the HW impact of rounding to nearest power? This seems significantly more complicated than adding uniform noise as it is the case for uniform quantization. "
>
> $\textbf{A5:}$ Uniform quantization requires a round-to-nearest operation. In round-to-nearest-power, as explained in eq. 19, we use the common round-to-nearest operation for the exponent. The only difference is the subtraction by the number (0.084), which is a very simple and commonly used operation.
>
> $\textbf{Q6:}$ "How do they define the pruning threshold? Are they learned/updated? I could not find any details on this in the paper "
>
> $\textbf{A6:}$ The "underflow threshold" in section 3 is updated in every bwd pass as part of the quantization process of the neural gradients according to the maximum of the tensor. We added a comment on this in section A.1.
>
> $\textbf{Q7:}$ " How do you deal with BN?"
>
> $\textbf{A7:}$ In LUQ, similar to previous methods ([1]), the BN is done in full-precision. We focus on reducing the GEMM operations, which takes the most part of the computational resources. We added a comment on this in section A.1.

---

> > ### Comment · Reviewer_na7b · 2021-11-22
> > **Thanks for the clarifications**
> >
> > Thank you for the detailed responds and clarifications. Here some follow up comments/questions:
> > * **Q1:** Thanks for pointing out the ablation study, I might have missed Fig 3a) as this was not referenced nor discussed in the text. What is not fully clear though is what the impact of stochastic pruning is, e.g. how would LUQ without SP train or how would RDNQ alone do? In the latest version it seems the figure is still not referenced, I would suggest the authors to add a reference and a short discussion for this ablation study.
> > * **Q6:** Updating the underflow value, would mean that in practice the full tensor needs to be known before quantizing the tensor. This means that, as pointed out by review 9cKC, this approach will significantly suffer from the memory transfer issues discussed in In-hindsight quantization.
> >
> > After carefully evaluating the other reviewers comments and considering the authors responds, I decided to keep for now my initial rating of 6. While the responds addressed some of my raised questions/concerns, there are also a few valid point raised by other reviewers which I did not notice in my initial review.
> >
> > One point that was less clear for me from the the initial manuscript is that the approach only tries to reduce the amount of GEMM operations which only partially utilizes the advantages of low bit quantization. It for example misses the opportunities to reduce memory transfer and saving area by not needing high precision compute cores.

---

> > > ### Author Response · Authors · 2021-11-23
> > > **Answer to clarifications of reviewer na7b**
> > >
> > > $\textbf{Q1:}$  "Q1 : Thanks for pointing out the ablation study, I might have missed Fig 3a) as this was not referenced nor discussed in the text. What is not fully clear though is what the impact of stochastic pruning is, e.g. how would LUQ without SP train or how would RDNQ alone do? In the latest version it seems the figure is still not referenced, I would suggest the authors to add a reference and a short discussion for this ablation study."
> > >
> > > $\textbf{A1:}$ We thank the reviewer for pointing out that we miss the reference for Fig 3(a). We uploaded a new revision where we added a reference for that figure. Next, we address reviewer questions. First, the effect of stochastic pruning (SP) can be observed in the ablation study of Fig 3(a): there, the standard FP4 that completely diverges, in contrast to FP4 + SP that is able to converge, though with some degradation. Second, following the reviewer request, in the new revision we added to Fig 3(a) an additional ablation study of FP4 + RDNP, showing it converges with significant degradation.  \\
> > >
> > > $\textbf{Q2:}$ " Q6: Updating the underflow value, would mean that in practice the full tensor needs to be known before quantizing the tensor. This means that, as pointed out by review 9cKC, this approach will significantly suffer from the memory transfer issues discussed in In-hindsight quantization. One point that was less clear for me from the the initial manuscript is that the approach only tries to reduce the amount of GEMM operations which only partially utilizes the advantages of low bit quantization. It for example misses the opportunities to reduce memory transfer and saving area by not needing high precision compute cores."
> > >
> > > $\textbf{A2:}$ Quantization in general is able to reduce 3 main parts of the resources needed in DNNs training: (1) computational resources by doing operations in low bits  (2) memory  (3) data transfer. When we need to change the dynamic range in quantization, it can be achieved by calculating statistics on the full tensor (as in LUQ, in Ultra-low [1] or in INT quantzation) or use known statistics (as in standard FP or in In-hindsight). We agree with the reviewer that the former requires more data transfer, than the second in part of the DNNs training process. However, it is not true to say LUQ utilizes only part of the advantages of quantization. LUQ reduces the computation resources (GEMM operations), the memory, and the data transfer (e.g., the activations that are saved for the bwd pass are in low bits).
> > > Notice, that as shown in Fig 5 (a), trying combine LUQ with In-hindsight quantization induces significant accuracy degradation. We believe this additional data transfer is not significant in comparison to the SOTA results achieved with LUQ on 4-bit training. As far as we know all the training accelerators (TPU - Google, GPU - Nvidia, Gaudi - Habana / Intel, IPU - Graphcore) are based on a mixed precision engine and are able to support multi datatypes, where usually bitwidth reduction increases linearly the throughput. Based on that we believe the need of high precision core is not a bottleneck in the design of training accelerators.

---

> > > > ### Comment · Area_Chair_BECa · 2021-12-01
> > > > **any additional thoughts or comments?**
> > > >
> > > > Thanks for this discussion! To the reviewer: any follow-up after reading this additional response from the authors?

---

> ### Author Response · Authors · 2021-11-15
> **Answer to reviewer na7b - Part 2**
>
> $\textbf{Q8:}$ "What quantization approach is applied in the forward pass (weights and activations)? Rounding to nearest is clear, but how are the ranges defined/learned etc. "
>
> $\textbf{A8:}$ As explained in section 4 in the "main results" paragraph we quantized the activations and weights to INT4 using a known method SAWB [2]. In SAWB the ranges are based on the statistical of the tensor (mean and std). As mentioned in the introduction, the main challenge to reduce the numerical precision in training is the quantization of the neural gradients and that is the problem that LUQ confronts. Quantization of the weights and activations was already well studied in previous papers where they were able to achieve 4-bit quantization with negligible degradation.
>
> ---------------------------------------------------------------------------------------------------------------------------------------------------------------------------
>
> [1] Xiao Sun, Naigang Wang, Chia-Yu Chen, Jiamin Ni, A. Agrawal, Xiaodong Cui, Swagath Venkatara-mani, K. E. Maghraoui, V. Srinivasan, and K. Gopalakrishnan. Ultra-low precision 4-bit training of deep neural networks. In NeurIPS, 2020
>
> [2] Jungwook Choi, P. Chuang, Zhuo Wang, Swagath Venkataramani, V. Srinivasan, and K. Gopalakrishnan. Bridging the accuracy gap for 2-bit quantized neural networks (qnn). ArXiv, abs/1807.06964,
> 2018a.

---

### Official Review · Reviewer_9cKC · 2021-11-04

**Correctness:** 2
**Technical Novelty And Significance:** 3
**Empirical Novelty And Significance:** 2
**Recommendation:** 5
**Confidence:** 4

**Main Review:**

This paper is interesting, but leaves me with too many questions and uncertainties on all three of it's contributions to make a proper judgment. It would be great if the authors could address my questions below for clarity:

I appreciate the rounding scheme analysis w.r.t. the MSE. I haven't seen that specifically before, and it's insightful to understand why stochastic rounding on the forward pass is not necessarily a good idea. There is one small caveat that should be mentioned, and that is that noise sometimes helps for proper convergence and regularization. Dropout adds noise, and increases the MSE, but still sometimes helps. It would be good if the authors rephrased this section slightly to take this into account properly.

The conclusion on page 3 is much too fast and needs to be worked out significantly more for clarity. Paraphrasing: "Unbiased gradients are necessary for convergence, therefore gradients should be quantized with stochastic rounding". I can't find the statements regarding the biased gradients in Bottou 2010, perhaps the authors could point me to where exactly this is mentioned.
I would also like to understand better what the bias of the gradients actually means, and why this would be so detrimental. If you talk about the bias of the forward pass given the stochastic distribution, I understand. However, it's unclear to me what the effect of this is on the gradients themselves. It would be good if the authors worked this out. What are the gradients biased with respect to? The FP32 gradients? Is that even the correct thing to compare to?
To me, figure 1 is insufficient evidence that a 'bias' is necessarily the issue in training. This effect would have to be disentangled from other effects that adding noise to a network would have on training. As it stands, the difference could come from just the addition of noise.
Similarly, The argument that RDN should be used in the forward pass goes much too fast for me as well. It would be nice if the authors showed explicitly how the linearity causes a non-biased estimate of the gradients in the forward pass.

Section 3.1 Extra sampling would also linearly increase the cost of the method. So why would this be better than simply training more iterations on different data as opposed to the same data? A discussion of the overhead/complexity of this method already here would be much appreciated. For the results section, I also don't see how the overhead comes into play with the comparison the raw LUQ method itself.

Section 3.2 Wouldn't we generally want the activations in a lower bit-width as well?

Method - It is unclear to me if the weights in this scheme are kept in a quantized format, or if they are in floating points with the common 'shadow-weight' approach. Please take more care in describing this properly. If the weights are in floating point, that would mean a very significant overhead during training, as the floating points would have to be loaded somewhere, then quantized, then an operation is to be applied to them. In full quantized training, you would want to have the weights quantized as well, but I see no mention of this. Similarly, are the updates to the weights quantized? See e.g. the WAGE paper.

Underflow Threshold - What are you taking the maximum over? If you are taking a maximum over all the values in the tensor you are quantizing, you will run into problems with quantization as described in the In-Hindsight quantization paper (https://arxiv.org/pdf/2105.04246.pdf). Essentially, for activation quantization, if you need any statistic of a tensor to do the quantization, you have to write significant amounts of data to memory, which is very slow. You might as well not have done activation/gradient quantization at that point.

Overhead of SMP and FNT. This number shouldn be 8x compared to FP16. Compared to FP32 it would be 16x at least.

Main results. It seems you are using at least a different ResNet-18 model compared to the Ultra-low paper, as the baseline accuracy is 69.7% for your model, but Ultra-low reports 69.4. Could some difference not be explained based on diffent models? The delta in performance is the same, roughly 0.7 for ResNet18. Also, how does this method compare to other works before it like WAGE (https://arxiv.org/pdf/1802.04680.pdf) and other papers that are based on this?

You mention that the shortcut connection in ResNets and the depth-wise layers are kept in full precision. For the depth-wise layer, does that mean the input activations to it, or the output activations? For both, I don't think this is necessarily trivial to do. It's not a given that a fixed point training engine also has a full-precision engine on it's die. Having that would significantly increase the die size. Similarly, although the compute is small of this solution, the data movement is not. Data movement is a very important metric on efficient devices (the ones we would be doing quantized training for). This needs to be addressed in the context of overall efficiency.

MF-BPROP - I definitely appreciate the hardware implementation, as if you'd have just the FP7 calculation MAC array, you might as well just do INT8 computations all the way as opposed to going throug the trouble of quantizing everything to INT4/FP4. But this section does highlight one problem with this method: Dedicated hardware is necessary for this method to work efficiently. This greatly limits the scope of this work in it's practicality. And this brings me back to the above data movement questions that were glossed over. The suggested method is not a general method for quantized training and more of a 'hardware and software go hand-in-hand' kind of method, but many of the practical details of the hardware implementation and costs/overhead are left out. Please comment on this.

Type-o's
page 3 round-to-zero | nearest
page 4 avoid of clipping | -of
Page 5 The a different samples | -a
Page 6 4-bit training in various DNN models | in -> on



**Summary Of The Paper:**

The paper has a short analysis on rounding schemes, comparing nearest-rounding to stochastic rounding for quantized training. It also introduces LUQ, a new quantized training scheme with a FP4 format for the gradients. In order to make this work efficiently in hardware, they introduce a new hardware block that does multiplication-less gradient calculations for the backward pass.

**Summary Of The Review:**

This is an interesting paper with some interesting insights, but the paper tries to do too much in too small a space. This leads me to have too many open questions for a good rating. The paper would likely have been better if it honed in on a single aspect that they present, and did so in a clearer fashion with more bases covered.
The conclusions on page 3 are drawn much too fast, and are unclear. The authors have to convince the reader a lot harder to make the claims they make.
On the one hand the paper's main method only seemingly work more efficiently with very dedicated hardware implementations, but several important aspects of efficient hardware implementations of training, such as data movement, are seemingly ignored.

I would be willing to increase my rating if the authors addressed my above questions, rewrote section 2 significantly with a more convincing analysis, and added a better complexity analysis of the overhead of their method in terms of data movement.

*** post-rebuttal ***

Given the author's excellent response to my questions, and the questions of the other authors, I am increasing my review to a 5. For a 6 or higher I definitely need the efficiency conundrum resolved, as for now I just don't see how to resolve the dichotomy between being either a generally applicable method on common-day and general hardware, or a method that is really aimed at hyper-efficient dedicated implementations.

---

> ### Author Response · Authors · 2021-11-15
> **Answer to reviewer 9cKC - Part 1**
>
> $\textbf{Q1:}$ "[...]There is one small caveat that should be mentioned, and that is that noise sometimes helps for proper convergence and regularization. Dropout adds noise, and increases the MSE, but still sometimes helps. It would be good if the authors rephrased this section slightly to take this into account properly"
>
> $\textbf{A1:}$ We thank the reviewer for his comment. We agree that in some cases, such as dropout or SGD noise in which noise helps to generalization. As suggested by the reviewer, we added this point in the new version of the conclusions of section 2.
>
> $\textbf{Q2a:}$  I can't find the statements regarding the biased gradients in Bottou 2010, perhaps the authors could point me to where exactly this is mentioned. [..]
>
> $\textbf{A2a:}$ Apologies, we accidentally had the wrong citation (with a similar name). The correct citation is "Optimization Methods for Large-Scale Machine Learning Léon Bottou, Frank E. Curtis, and Jorge Nocedal SIAM Review 2018 60:2, 223-311"
> Please see assumption 4.3(b) and the discussion below it (page 23 of the pdf) which clarifies why unbiased estimates satisfy this assumption. This assumption is used in the convergence results later (e.g., Theorem 4.8-4.10 in the non-convex case). We corrected this in the revised paper.
>
> $\textbf{Q2b:}$ What are the gradients biased with respect to? The FP32 gradients? Is that even the correct thing to compare to? [...]  It would be nice if the authors showed explicitly how the linearity causes a non-biased estimate of the gradients in the forward pass.
>
> $\textbf{A2b:}$ Quantized gradients are biased with respect to the real-valued gradients. Following the reviewer's suggestion, we edited section 2. Specifically, we added section 2.1, where we showed that the unbiasedness at the tensor level in the bwd pass induced unbiasedness at the model level, which is different in the forward pass since the non-linear activations break the unbiasedness.
>
> $\textbf{Q3:}$ "Extra sampling would also linearly increase the cost of the method. So why would this be better than simply training more iterations on different data as opposed to the same data? A discussion of the overhead/complexity of this method already here would be much appreciated. For the results section, I also don't see how the overhead comes into play with the comparison the raw LUQ method itself."
>
> $\textbf{A3:}$ The main purpose of LUQ is to create an unbiased quantization for the neural gradients. After solving the bias problem, we wanted a method to reduce the variance without affecting the unbiasedness of LUQ. There are many ways to reduce the variance without affecting the bias - in this paper, we focus on sampling as presented in section 3.1. In Figure 3b the reviewer can notice the effect of the sampling on the accuracy, which is able to close the gap to the baseline with 16 samples on 2 bits quantization in Cifar100 dataset. Additionally, we present in the experiments section the accuracy improvement in various models for ImageNet dataset (e.g 0.39 \% in ResNet18), where we limit our experiment to only one additional sample to be comparable in the overhead with the TPR method presented in [1]. To improve the overhead discussion we edited section 3.1. Now, we emphasize that the power overhead is around 1/3 of the number of additional samples since it affects only the update GEMM. The throughput overhead is minor since the different samples can run in parallel. Following the reviewer's suggestion, we added in Fig 5b an experiment that compares the SMP method with longer training to get comparable overhead. As the reviewer can notice, the SMP method leads to better accuracy.

---

> ### Author Response · Authors · 2021-11-15
> **Answer to reviewer 9cKC - Part 2**
>
> $\textbf{Q4:}$ "Section 3.2 Wouldn't we generally want the activations in a lower bit-width as well? It is unclear to me if the weights in this scheme are kept in a quantized format, or if they are in floating points with the common 'shadow-weight' approach. Please take more care in describing this properly. If the weights are in floating point, that would mean a very significant overhead during training, as the floating points would have to be loaded somewhere, then quantized, then an operation is to be applied to them. In full quantized training, you would want to have the weights quantized as well, but I see no mention of this. Similarly, are the updates to the weights quantized? See e.g. the WAGE paper "
>
> $\textbf{A4:}$ In sec 3.2 we presented a method in which, after training the network in low bit-width, we train for only 1 additional epoch, where all the network increase to full precision, except the weights, which are trained in low precision. Then for inference, the activations are lowered back to low bits.  We noticed empirically that this scheme gets the best results. We improved the explanation of this method in section 3.2. In respect to the reviewer's another comment, our work follows the most common approach where, as the reviewer described, a "shadow" weight copy is saved in high-precision and quantized on-the-fly. Weight updates are done in full precision using quantized gradients. Training without high-precision updates (as demonstrated in WAGE paper) is an additional challenge with an accuracy penalty on its own. We make these details clearer in sec A.1.
>
> $\textbf{Q5:}$ "Underflow Threshold - What are you taking the maximum over? If you are taking a maximum over all the values in the tensor you are quantizing, you will run into problems with quantization as described in the In-Hindsight quantization paper (https://arxiv.org/pdf/2105.04246.pdf). Essentially, for activation quantization, if you need any statistic of a tensor to do the quantization, you have to write significant amounts of data to memory, which is very slow. You might as well not have done activation/gradient quantization at that point. "
>
> $\textbf{A5:}$ In LUQ the underflow threshold calculation requires the maximum over all values in the tensor. We thank the reviewer for mentioning the In-Hindsight quantization paper which shows comparable results in 8-bit training while reducing the data movement overhead in the calculation of the quantization dynamic range. We added it to the related work list in the new revision. In order to check the effect of the dynamic range approximation, we added in the new revision Fig 5(a) an experiment that compare LUQ with the maximum approximation method proposed in In-Hindsight. As the reviewer can notice, this approximation induces a significant accuracy degradation. Unfortunately, we believe that the limited dynamic range in 4-bit quantization does not work well with such approximations, and requires an exact measurement to remain unbiased. This is in contrast to 8-bit quantization, where such an approximation can work well. We agree with the reviewer that the data movement increase when the quantization requires any statistic of the tensor, as in LUQ or in standard fixed-point quantization. However, as shown in our experiments it's a necessary price to pay to achieve better accuracy. We thanks the reviewer for pointing out this interesting future research direction to reproduce LUQ results with less data movement.
>
> $\textbf{Q6:}$ "Overhead of SMP and FNT. This number shouldn be 8x compared to FP16. Compared to FP32 it would be 16x at least."
>
> $\textbf{A6:}$ The reviewer is right. The overhead compared to FP16 is 8x (as explained in [1] appendix-C) and therefore 16x compared to FP32. We fix it in the new revision.
>
> $\textbf{Q7:}$ "Main results. It seems you are using at least a different ResNet-18 model compared to the Ultra-low paper, as the baseline accuracy is 69.7\% for your model, but Ultra-low reports 69.4. Could some difference not be explained based on diffent models? The delta in performance is the same, roughly 0.7 for ResNet18. Also, how does this method compare to other works before it like WAGE  and other papers that are based on this?"
>
> $\textbf{A7:}$ We don't know why the ResNet-18 baseline of Ultra-low [1] is 69.4 \%. We believe it's a typo since it is lower than their results where they quantize the fwd to INT4 and the bwd is kept in fp32. They don't share the code (we tried to contact the authors), so we cannot reproduce their experiments. Our baseline is similar to the torchvision baseline and the code is supplied in the supplementary material. Moreover, even with their low baseline, our delta is still smaller (0.7 \%  Vs 1.13 \%). WAGE quantized the activations and neural gradients to 8 bits. We compared our work only with Ultra-low since as we know it's the only work that quantized the weights, activations and neural gradients to 4-bits.

---

> ### Author Response · Authors · 2021-11-15
> **Answer to reviewer 9cKC - Part 3**
>
> $\textbf{Q8:}$ "You mention that the shortcut connection in ResNets and the depth-wise layers are kept in full precision. For the depth-wise layer, does that mean the input activations to it, or the output activations? For both, I don't think this is necessarily trivial to do. It's not a given that a fixed point training engine also has a full-precision engine on it's die. Having that would significantly increase the die size. Similarly, although the compute is small of this solution, the data movement is not. Data movement is a very important metric on efficient devices (the ones we would be doing quantized training for). This needs to be addressed in the context of overall efficiency."
>
> $\textbf{A8:}$ For the depth-wise layer we mean the input activations to it. As far as we know all the training accelerators (TPU - Google, GPU - Nvidia, Gaudi - Habana / Intel, IPU - Graphcore) are based on a mixed precision engine and are able to support multi datatypes, where usually bitwidth reduction increases linearly the throughput [4]. The purpose of our work was mainly to reduce the computational GEMM overhead. We emphasize this in the introduction in the new revision. This is also the most common metric used in low-bitwidth training research. We agree with the reviewer that data movement can be also an important metric and an interesting research direction. It can be reduced in hardware such as Graphcore, with their in-processor-memory, in the algorithm, such as the In-Hindsight paper mentioned above.
>
> $\textbf{Q9:}$ "MF-BPROP - I definitely appreciate the hardware implementation, as if you'd have just the FP7 calculation MAC array, you might as well just do INT8 computations all the way as opposed to going throug the trouble of quantizing everything to INT4/FP4. But this section does highlight one problem with this method: Dedicated hardware is necessary for this method to work efficiently. This greatly limits the scope of this work in it's practicality. And this brings me back to the above data movement questions that were glossed over. The suggested method is not a general method for quantized training and more of a 'hardware and software go hand-in-hand' kind of method, but many of the practical details of the hardware implementation and costs/overhead are left out. Please comment on this"
>
> $\textbf{A9:}$. Our work contains 2 main and orthogonal topics. First is LUQ to quantize the neural gradient to FP4 format 1-3-0 (sign-exponent-mantissa) without bias in order to reduce the computational resources in DNNs training. Second is the hardware implementation suggestion called MF-BPROP which is a general hardware implementation that can avoid multiplications when one argument has only mantissa (like INT4 in our case) and the other argument only exponent (like FP4 1-3-0 in our case). Both methods can work together or separately, which means we can run LUQ with the standard FP engine without requiring dedicated hardware and also use MF-BPROP with additional quantization datatypes. We agree with the reviewer that the implementation of MF-BPROP require a specific hardware implementation but it comes with a significant logical area reduction as explained in sec A.4
>
> --------------------------------------------------------------------------------------------------------------------------------
>
> [1] Xiao Sun, Naigang Wang, Chia-Yu Chen, Jiamin Ni, A. Agrawal, Xiaodong Cui, Swagath Venkatara-mani, K. E. Maghraoui, V. Srinivasan, and K. Gopalakrishnan. Ultra-low precision 4-bit training of deep neural networks. In NeurIPS, 2020
>
> [4] https://www.nvidia.com/content/dam/en-zz/Solutions/Data-Center/a100/pdf/nvidia-a100-datasheet.pdf

---

> ### Comment · Reviewer_9cKC · 2021-11-29
> **Rebuttal response**
>
> All comments are greatly appreciated. It's good to see the authors are taking good care of the comments of all reviewers, and have good input/comments on all of them. The rebuttal is exemplary for other authors in my opinion. Very well done.
>
> > A 2b
>
> Thanks for the clarification on the citation. I fully understand the effect of bias on the backward-pass, as described in the new section 2.1. This is quite logical, as e.g. a small gradient could always be rounded down to 0 carrying no information, the algorithm wouldn't converge, whereas a unbiased stochastic rounding would at least on average nudge the weight value in the right direction.
> I still don't understand this part in the document however:
> "The forward pass is different from the backward pass in that unbiasedness at the tensor level is not
> necessarily a guarantee of unbiasedness at the model level since the activation functions and loss
> function are not linear. Therefore, even after stochastic quantization, the forward phase remains
> biased." I would still like to see this expanded better (the appendix will also do for this)
>
> > A 3
>
> I see what you are doing here. I do question it's efficacy however. Wouldn't this mean you had to store all intermediate feature maps in memory? That is, similar to the other memory comments I made, quite problematic on power-efficient target devices. But even ignoring the memory component, if you're doing 16 updates with your FP4/INT2 combined scheme, wouldn't you be better off doing a single INT8 or FP16 update? I don't see how multiple of these INT4 updated samples that cost 1/3rd of a full pass couldn't just have been a higher bit-width update instead. This is still quite unclear to me.
>
> > A 5
> I greatly appreciate the comments. However, this to me is a certain weak point for the method. The moment you have to write an FP32 tensor to memory, then read it back in to take a maximum in some form of tensor/vector core, then write it out again, only to read it back in again to quantize it down to lower bit-widths gives a very significant amount of FP32 data reading overhead. This is slow, and goes against the efficient on-device use-case setup of this paper.
>
> > A 8
> Intel is moving to a separated integer core, and floating point core, specifically because mixed-precision cores are much slower for both tasks, there was a paper on this recently. Several other vendors of efficient devices, such as TPUv3/v4 and many phones also have split FP and INT cores. Although you wouldn't be able to tell from their descriptions and APIs directly, but it's implemented that way in the hardware. More on the contents of this point below.
>
> > A 9
>
> This point is crucial, and prevents me from giving a higher rating than the adjustments I will make below. As I see it, based on the paper and the author's explanation, either of the following two scenario's hold:
> - You don't have the dedicated hardware implementation, and we're using commonly used hardware as mentioned by the authors in Q/A8. On top of the extra FP32 data movement, the INT4/FP4 scheme needs a FP7 calculation unit. If this is the case, in this scenario I'd rather have a INT8 calculation unit, which is just as fast. The FP7 calculation unit is not standard on those commonly used hardware devices anyway. I assume most of these devices do not have your specific FP scheme as well. Factoring in the extra data movement due to the reasons above, the main LUQ scheme for INT4/FP4 training does not seem to give me any benefits over INT8 training.
> - You do have dedicated hardware. This means you are likely going for a super-efficient on-device implementation. However, in these scenarios you'd very likely rather reduce the significant amount of FP32 data movement overhead you would have during training, before optimizing the bit-widths from something 8-bitty to 4 bits. The extra data movement would be incredibly killing in these hyper-efficient scenarios.
>
> Hence in both-scenarios, the method seems lacking in practicality to me.
>
> Given the author's excellent response to my questions, and the questions of the other authors, I am increasing my review to a 5. For a 6 or higher I definitely need the above conundrum resolved, as for now I just don't see how to resolve the dichotomy between being either a generally applicable method on common-day and general hardware, or a method that is really aimed at hyper-efficient dedicated implementations.

---

> ### Author Response · Authors · 2021-11-30
> **Answer to the rebuttal response of reviewer 9cKC - Part 1/2**
>
>
> We thank the reviewer for the positive and constrictive feedback, and for increasing the score. See below our response for the additional comments:
>
>
> $\textbf{Q1:}$ "A2b [...] The forward pass is different from the backward pass in that unbiasedness at the tensor level is not necessarily a guarantee of unbiasedness at the model level since the activation functions and loss function are not linear. Therefore, even after stochastic quantization, the forward phase remains biased." I would still like to see this expanded better (the appendix will also do for this)"
>
>
> $\textbf{A1:}$ In this part we emphasize that  the non-linearity of the activations (and the loss) anyway creates a bias that cannot be fixed with an unbiased quantization. In other words, when using an unbiased quantization in the forward path, the non-linear function will create a bias in the next layer. For example, suppose we have two layers with weights $W_1, W_2$, activation $\sigma$, input $x$, the SR quantizer $Q$, and denote as $\mathbb{E}$ the expectation with respect to the randomness in $Q$. Then, despite that $Q$ is unbiased (i.e., $\mathbb{E}[Q(x)]=x$), we get
>
>  $\[\mathbb{E}[ \sigma (W_2 Q (\sigma ( W_1 x))]  \neq  \mathbb{E}[ \sigma (W_2  (\sigma ( W_1 x))] \]$
>  since $\sigma$ is non-linear.
>  This means there is no point to use SR in the forward path, since it will increase the MSE (from eq. 9), but it will not fix the bias issue. In contrast, the backward pass (as represented in eq. 13) is linear, so the unbiased quantization of the neural gradient (LUQ) will create an unbiased update (as shown in eq. 14).  The conclusion of this analysis is that the forward path it is not recommended to use SR, but SR is recommended in the backward path. Following the reviewer request, we will clarify this point in the final version of the manuscript.
>
>
>
> $\textbf{Q2:}$ " A3: Wouldn't this mean you had to store all intermediate feature maps in memory? That is, similar to the other memory comments I made, quite problematic on power-efficient target devices. But even ignoring the memory component, if you're doing 16 updates with your FP4/INT2 combined scheme, wouldn't you be better off doing a single INT8 or FP16 update? I don't see how multiple of these INT4 updated samples that cost 1/3rd of a full pass couldn't just have been a higher bit-width update instead. This is still quite unclear to me."
>
> $\textbf{A2:}$ We agree with the reviewer that, in terms of power, using multiple samples can be comparable to increasing the bandwidth. However, since this only affects the update phase, it can done in parallel. This means that we are still able to get significant improvements using low bit computations, in terms of throughput.
>
> $\textbf{Q3:}$ "A8: Intel is moving to a separated integer core, and floating point core, specifically because mixed-precision cores are much slower for both tasks, there was a paper on this recently. Several other vendors of efficient devices, such as TPUv3/v4 and many phones also have split FP and INT cores. Although you wouldn't be able to tell from their descriptions and APIs directly, but it's implemented that way in the hardware. More on the contents of this point below"
>
> $\textbf{Q3:}$ We thank the reviewer for pointing out the trend in mixed precision computing. However, we wish to point out this is not the case for all hardware solutions. For example, NVIDIA's tensor-cores still employs mixed-precision computations using the same hardware units.

---

> ### Author Response · Authors · 2021-11-30
> **Answer to the rebuttal response of reviewer 9cKC - Part 2/2**
>
> $\textbf{Q4:}$ "A5: I greatly appreciate the comments. However, this to me is a certain weak point for the method. The moment you have to write an FP32 tensor to memory, then read it back in to take a maximum in some form of tensor/vector core, then write it out again, only to read it back in again to quantize it down to lower bit-widths gives a very significant amount of FP32 data reading overhead. This is slow, and goes against the efficient on-device use-case setup of this paper."
>
> $\textbf{A4:}$ We agree with the reviewer that reducing the data movement is an interesting research topic which we believe can improve and benefit our method further. Please look at the answer $A5$ for additional details in respect to the data movement bottleneck.
>
>
> $\textbf{Q5:}$ "A9: [..]  As I see it, based on the paper and the author's explanation, either of the following two scenario's hold:[..]"
> $\textbf{A5:}$ In respect to the first bullet, current hardware such as NVIDIA GPU can use the same tensor core for lower bitwidth calculation and get a linear improvement in the throughput, instead of quadratic as expected in a dedicated hardware. For example they are able to get 2x improvement in INT8 in respect to FP16 (and not 4x as expected for a dedicated hardware). Using this scenario we still see the throughput benefit of our method which can used on commonly used hardware.
> However, our method is more relevant for the second bullet, of having a dedicated hardware. We agree with the reviewer that data movement in high-precision may be one of the bottlenecks that needs to be addressed when planning such hardware. Our work focused on resolving another bottleneck, which is the 4-bit GEMM operations, with SOTA results. We thank the reviewer for pointing out that the data-movement bottleneck may become more dominant after our work alleviated the GEMM bottleneck. Since this data movement bottleneck is now highlighted, we believe future work would invest much more effort in solving it. This can potentially be done in software, with methods which sub-sample the tensor, reduce the bitwidth, or improve the method suggested by In-Hindsight. In addition, this issue might be solved with dedicated hardware such as a unit that calculates the maximum value more efficiently or use memory-on-chip blocks which reduce data-movement overhead.  We want to emphasize that we are fully comparable in term of data movement with previous papers such as Ultra-Low [1] (Oral presentation - Neurips 2020), which did not mention the data movement bottleneck. We leave this issue stemming from required measurements and their possible overhead as a focus point for future work. Anyway, we thank again the reviewer for pointing out this data movement problem. We plan to clarify this point and possible solutions in the final version of the manuscript.

---

### Author Response · Authors · 2021-11-15
**General comment**

We thank all the reviewers for their helpful feedback and remarks. We uploaded a new revision of the paper to address these remarks.
Notice the additional experiment in a Transformer-base model in Table 2, where LUQ alone (i.e., even without SMP and FNT) achieves less than 0.4 BLUE score degradation, in comparison to 2.1 BLUE score degradation in Ultra-low [1].
Additionally, we fix all the typos mentioned by the reviewers. For additional details, see the answer to each reviewer's concerns. Please let us know if there are any additional comments.


[1] Xiao Sun, Naigang Wang, Chia-Yu Chen, Jiamin Ni, A. Agrawal, Xiaodong Cui, Swagath Venkatara-mani, K. E. Maghraoui, V. Srinivasan, and K. Gopalakrishnan. Ultra-low precision 4-bit training of deep neural networks. In NeurIPS, 2020

---

### Author Response · Authors · 2021-12-02
**General comment**

Before this discussion ends, we would like to thank all the reviewers for their positive and helpful feedback and to summarize our current perspective. As indicated by the reviewers' responses so far, we believe we have clarified all the reviewers' concerns, except one: the issue of data movement, raised by reviewer 9cKC.
Despite its importance of data movement, we feel this issue is outside of the scope of this paper for two reasons:
1) This is an issue that is highly hardware-dependent, and in some devices, it is not a serious issue (e.g., Graphcore with their memory-on-chip device). Moreover, given our paper addresses the GEMM bottleneck, one can now focus efforts on designing devices that reduce the data-movement issue.
2) Almost all previous algorithmic papers on quantization did not measure or address data movement at all, including very successful ones (e.g., Ultra-Low [1], a NeurIPS 2020 oral presentation).

Our paper is the first to show that all GEMM operations can be practically done in 4-bit, and we strongly believe this is sufficient to warrant a publication.

---------------------------------------------------------------------------
[1] Xiao Sun, Naigang Wang, Chia-Yu Chen, Jiamin Ni, A. Agrawal, Xiaodong Cui, Swagath Venkatara-mani, K. E. Maghraoui, V. Srinivasan, and K. Gopalakrishnan. Ultra-low precision 4-bit training of deep neural networks. In NeurIPS, 2020

---

### Decision · Program_Chairs · 2022-01-20

**Decision:**

Reject

**Comment:**

This paper proposes a method for 4-bit quantized training of NNs (forward and backward), obtaining SOTA 4-bit training quantization, motivated by an analysis of rounding schemes (an important aspect) in quantized training. The main concerns from the reviewers were that the approach was not practical (both a general concern, and of specific note here since the word is used in the title and motivation of the work), due to lack of compatibility with (current) general purpose hardware, and lack of suitability of the approach for specialized hardware, so it is unclear what the actual use case is for the approach. The authors argued that (1) this is not a problem on some hardware and (2) that past works have not been held to this standard. I did not find the authors to provide a strong argument during the discussion period to address these concerns.